# Prognostic and Predictive Roles of HER2 Status in Non-Breast and Non-Gastroesophageal Carcinomas

**DOI:** 10.3390/cancers16183145

**Published:** 2024-09-13

**Authors:** Erica Quaquarini, Federica Grillo, Lorenzo Gervaso, Giovanni Arpa, Nicola Fazio, Alessandro Vanoli, Paola Parente

**Affiliations:** 1Medical Oncology Unit of Pavia Institute, Istituti Clinici Scientifici Maugeri IRCCS, 27100 Pavia, Italy; erica.quaquarini@gmail.com; 2Anatomic Pathology Unit, University of Genova and Policlinico San Martino Hospital, 16132 Genova, Italy; federica.grillo@unige.it; 3Division of Gastrointestinal Medical Oncology and Neuroendocrine Tumors, European Institute of Oncology, IRCCS, 20141 Milan, Italy; lorenzo.gervaso@ieo.it (L.G.); nicola.fazio@ieo.it (N.F.); 4Department of Molecular Medicine, University of Pavia, 27100 Pavia, Italy; giovanni.arpa@icsmaugeri.it; 5Anatomic Pathology Unit of Pavia Institute, Istituti Clinici Scientifici Maugeri IRCCS, 27100 Pavia, Italy; 6Anatomic Pathology Unit, Fondazione IRCCS San Matteo Hospital, 27100 Pavia, Italy; 7Pathology Unit, Fondazione IRCCS Casa Sollievo della Sofferenza, 71013 San Giovanni Rotondo, Italy; paolaparente77@gmail.com

**Keywords:** *ERBB2* gene, anti-HER2 therapies, HER2 overexpression, *HER2* amplification, *HER2* mutation, epithelial tumor

## Abstract

**Simple Summary:**

Over the past years, the introduction of various anti-HER2 therapies has significantly improved the outcome for patients with HER2-positive breast and gastroesophageal carcinomas. HER2 protein overexpression is investigated using immunohistochemistry, gene amplification using fluorescence in situ hybridization, and gene mutation using next-generation sequencing. This review evaluated the predictive and prognostic role of HER2 status in various types of epithelial malignant cancers beyond breast and gastroesophageal cancers, focusing on the published studies, the scoring systems, and assays used and analyzing the clinical parameters and therapeutic approaches used. The evidence about prognostic and predictive roles of HER2 in carcinomas other than breast and gastroesophageal remains investigational but is increasing due to a tumor site-related prognostic and predictive value of the different types of HER2 alterations. The major limitation was that standardized and validated scoring system assays are not well-established for many organs.

**Abstract:**

The oncogene *ERBB2*, also known as *HER2* or *c-ERB2*, is located on chromosome 17 (q12). It encodes a tyrosine kinase receptor, the human epidermal growth factor receptor 2 (HER2), involved in neoplastic proliferation, tumor angiogenesis, and invasiveness. Over the past years, the introduction of various anti-HER2 therapies has significantly improved outcomes for patients with HER2-positive breast and gastroesophageal carcinomas. More recently, the introduction of a new antibody–drug conjugate, that is trastuzumab deruxtecan, expanded the therapeutic options to low-HER2 breast and gastroesophageal tumors. HER2 protein overexpression is investigated using immunohistochemistry, gene amplification using fluorescence in situ hybridization, and gene mutation using next-generation sequencing. This review evaluated the predictive and prognostic role of HER2 status in various types of epithelial malignant cancers beyond breast and gastroesophageal cancers. We critically analyzed the key published studies, focusing on utilized scoring systems and assays used, and analyzed clinical parameters and therapeutic approaches. Although the evidence about prognostic and predictive roles of HER2 in carcinomas other than breast and gastroesophageal has been widely increasing over the last decade, it still remains investigational, revealing a tumor site-related prognostic and predictive value of the different types of HER2 alterations. However, standardized and validated scoring system assays have not been well-established for many organs.

## 1. Introduction

The oncogene *ERBB2* (HUGO Gene Nomenclature Committee ID: 3430, name: erb-b2 receptor tyrosine kinase 2), also known as *HER2* or *c-ERB2,* is located on chromosome 17 (q12). This gene encodes for a tyrosine kinase receptor called erbB-2 or human epidermal growth factor receptor 2 (HER2). HER2 activation leads to the stimulation of oncogenic pathways, resulting in increased cell proliferation, tumor angiogenesis, and invasiveness. [1]. Out of all four proteins in the HER family, HER2 has the highest and strongest catalytic kinase activity and induces the most active signaling after combining with other HER family members through homodimerization (HER2:HER2) or heterodimerization (such as HER2:HER3). Protein expression levels of HER2 are determined using immunohistochemistry (IHC), while gene amplification is measured using fluorescence in situ hybridization (FISH). In FISH analysis, HER2 overexpression is usually considered present if the *HER2*:CEP17 ratio is equal to or greater than 2.0. Validated scoring systems for assessing HER2 protein expression have been developed for breast carcinoma, gastroesophageal carcinoma, and colorectal cancer [2,3,4]. Over the past few years, the development of various anti-HER2 treatments has significantly improved treatment outcomes for patients with HER2-positive carcinomas. These improvements vary depending on the site of origin and histology of the tumor. Moreover, the introduction of a novel type of treatment, called antibody–drug conjugates (ADCs), particularly trastuzumab deruxtecan (T-DXd), has expanded the range of therapeutic options for HER2-low breast and gastric carcinomas [5]. ADCs consist of a high-activity cytotoxic payload conjugated to an anti-HER2 monoclonal antibody that accurately delivers cytotoxic drugs into tumor cells. Overall, these developments have contributed to the progress in oncologic outcomes for patients. Moreover, in the last decade, HER2 expression has been investigated in non-breast and non-gastroesophageal carcinoma, showing a possible influence on carcinogenesis and a promising role as a target for oncologic treatment. However, no codified scoring and validated assays for HER2 evaluation exist for all tumor sites. Furthermore, the genomic profiling of human cancers has discovered recurrent somatic mutations in *ERBB2*, which can occur even without gene amplifications [6]. An analysis of sequencing data from approximately 111,000 tumors, encompassing around 400 types or subtypes of cancer, has found numerous recurring somatic point mutations in the transmembrane and juxtamembrane domains of HER2, resulting in the substitutions of amino acids. Functional investigation of these recurring mutations has indicated that a majority of them are activating and are likely to be key alterations driving cancer development. Unlike other cancer-causing genes, no single prevalent mutant *ERBB2* allele has been found, and the distribution of mutations differs depending on the type of tumor. Somatic *ERBB2* alterations are typically rare, occurring in only a small percentage of cancer cases (1–3%) across various types of cancer. These alterations can sometimes be present alongside *HER2* gene amplification. *HER2* mutations, such as V841I, S310F, L755S, V777L, and I655V, affect different regions of the ERBB2 protein, causing continuous activation of signals that promote cell growth [6].

In this review, we evaluated the predictive and prognostic role of HER2 status (including overexpression, amplification, and mutation) in various types of carcinoma beyond breast and gastroesophageal tumors described in the most important studies in the literature, with special regard to the scoring and assays used, the clinical parameters analyzed, and therapeutic approaches.

## 2. HER2 Role in Different Types of Epithelial Tumors

### 2.1. Salivary Gland Carcinoma

#### 2.1.1. Epidemiology and Frequency of HER2 Alterations

Salivary gland carcinomas (SGCs) are rare and aggressive neoplasms, representing the 27th cancer for incidence and 28th for cancer mortality in the world [7]. They include numerous histotypes, each with peculiar molecular pathways. *HER2* amplification has been reported in 12–52% of SGC, HER2 overexpression in 17–44% of cases, and *HER2* mutations in only 1% [8] (Figure 1).

A recent meta-analysis estimated the prevalence of HER2 overexpression in all histologic subtypes of SGC [9]. HER2 expression was found in 43% of salivary duct carcinomas (SDCs), 39% of carcinoma ex pleomorphic adenomas (CEPs), 17% of squamous cell carcinomas (SCCs), 13% of adenocarcinomas not otherwise specified (ADCs NOS), 6.7% of poorly differentiated carcinomas, 5.5% of mucoepidermoid carcinomas, 4.3% of myoepithelial carcinomas, 1.8% of epithelial-myoepithelial carcinomas, 0.45% of acinic cell carcinomas, and 0.15% of adenoid cystic carcinomas.

#### 2.1.2. HER2 Evaluation Criteria

Although specific criteria for SGC have been proposed, the breast cancer carcinoma criteria for the evaluation of HER2 protein overexpression are the most commonly used for scoring HER2 expression in SGC [10] (Table 1). This is partly attributed to the morphological similarities between SGC and invasive ductal carcinoma of the breast, as well as the molecular resemblance to apocrine breast cancer. Additionally, there is a lack of studies validating HER2 scoring systems specifically for SGC.

#### 2.1.3. Prognostic Role of HER2 Alterations and Association with Clinicopathologic Features

HER2 overexpression is associated with worse outcomes in SDCs and AR-positive SGCs with a higher risk of recurrence [9]. By contrast, the HER2-related prognosis for ADCs NOS seems unclear. The overexpression of HER2 is typically associated with more aggressive tumors, a higher rate of distant metastasis and poor prognosis. Apocrine HER2 subtype (AR+/HER2+) of SGCs is significantly associated with lower overall survival. OS. With conventional chemotherapy, HER2-positive SDC has a high incidence of recurrence and rapid disease progression, regardless of tumor site, size, grade, and lymph node status. A recent study by Cavalieri and colleagues described the natural history of AR-positive recurrent/metastatic SGC patients based on *HER2* amplification status [11]. In this study, patients with HER2-positive disease had a statistically significant higher risk of disease recurrence or death from recurrent or metastatic disease. Moreover, the HER2-positive group showed a non-significant trend toward a higher prevalence of brain metastases with consequent reduced patient survival. Conversely, in other studies, such an association was not found; in fact, HER2 positivity did not impact disease-free survival (DFS) or OS among patients with SGCs [9]. All these findings together confirm the importance of assessing HER2 status at diagnosis of SDC and ADC NOS, at least. In the literature, there are no definitive data regarding the prognostic role of *HER2* mutations in this rare tumor type.

#### 2.1.4. Predictive Role of HER2 Alterations and Clinical Trials

Some reports have evaluated the potential of targeted agents for SGC with the clinical benefit of anti-HER2 drugs, particularly when used in an early line of treatment [12,13,14,15,16,17,18,19,20,21,22,23,24] (Table 2). By considering fully published papers including at least four patients, 13 studies were found, of which seven were retrospective, one was a phase I, and five were phase II studies. In nine of them, the anti-HER2 agent was used in advanced treatment lines, in five cases, in the first-line setting, and in two, also in the adjuvant setting. In most cases, HER2 expression was assessed with IHC using breast cancer criteria, with the subsequent FISH confirmation of amplification in cases of IHC 2+. Only the DESTINY-PanTumor02 trial used gastric cancer criteria to evaluate HER2 IHC/FISH. In 8/12 studies, trastuzumab was administered in combination with chemotherapy; in 2/12 studies, only trastuzumab was administered; in one study, trastuzumab *plus* pertuzumab was administered; in one study, ado-trastuzumab emtansine (T-DM1) was administered; and, in one study, T-DXd was administered. In about two-thirds of the studies, the primary endpoint was the activity of the treatment in terms of overall response rate (ORR), which reached a value of 70% in three studies [17,20,23]. Considering the advanced disease lines of treatment, survival data showed encouraging results in terms of progression-free survival (PFS) and OS. Trials with T-DM1 in combination with radiotherapy or chemotherapy, HER2-specific CAR-T, T-DXd, and trastuzumab/pertuzumab are currently ongoing [25]. No data regarding the predictive role of *HER2* mutations are present in the literature.

### 2.2. Head and Neck Carcinoma

#### 2.2.1. Epidemiology and Frequency of HER2 Alterations

Head and neck squamous cell carcinomas (HNSCCs) are aggressive neoplasms, representing the 23rd cancer for incidence and the 20th for cancer mortality in the world [7]. HER2 overexpression is often independent of gene amplification, leading to a lower overall level of expression than other tumors, with reported HER2 overexpression ranging from 0% to 88% [48] (Figure 1). No data regarding *HER2* amplifications and mutations are present in the literature.

#### 2.2.2. HER2 Evaluation Criteria

Since no standardized evaluation methods have been developed for HNSCC, IHC/FISH guidelines for breast carcinomas are typically applied to HNSCC despite their different etiology, biology, and clinical features (Table 1). A review of the studies present in the literature reveals variations in detection techniques (including the use of different antibodies) and interpretation methods, which contribute to varying reports of HER2 overexpression in this context. Moreover, no threshold for HER2 positivity has been definitively established.

#### 2.2.3. Prognostic Role of HER2 Alterations and Association with Clinicopathologic Features

HNSCC has an overall 5-year survival rate of 40–60%; TNM staging, pathologic grade of differentiation, and other clinical features often fail to adequately predict treatment response and survival [49]. A recent retrospective trial showed that 19% of HNSCC and 39% of oropharyngeal HNSCC were HER2-positive [48]. HER2 expression positively correlated with nodal metastases, while patients with HER2-positive tumors had decreased OS, including those with human papillomavirus (HPV)-positive tumors. In particular, HER2 expression and its correlation with survival seems to vary across HNSCC subsites, making it unsuitable as a prognostic marker.

#### 2.2.4. Predictive Role of HER2 Alterations and Clinical Trials

Recently, novel anti-HER2 agents have been tested in this context, but the results are still inconclusive regarding the possible use of these drugs in the context of advanced HER2-positive HNSCC (Table 2). For this reason, a predictive role of HER2 alterations has not been described in this context.

A large variety of antibodies, ADCs, and small molecule inhibitors of HER2 and HER3 signaling have been under study in early-stage clinical trials of HNSCC. Additional preclinical and early-stage clinical trials utilizing anti-HER2 and anti-HER3 therapies in combination with other targeted treatments (e.g., anti-vascular endothelial growth factor, pan receptor tyrosine kinase inhibitors) have been reported [49].

### 2.3. Lung Cancer

#### 2.3.1. Epidemiology and Frequency of HER2 Alterations

Lung cancer (LC) is the most frequent cancer and the first cause of mortality in the world, with 12.4% incidence and 18.7% mortality; it is the most frequent cancer diagnosed in men and the second in women [7]. Differences in tobacco exposure and air pollution reflect different geographic incidences and different prevalences in histotype, i.e., non-small cell LC and subtypes (NSCLC, squamous and adenocarcinoma) vs. small cell lung cancer (SCLC) [50]. Target therapy is suitable only for a small cohort of patients, raising the need for new strategies. In this context, *HER2* gene amplification, protein overexpression, and mutations represent an interesting area of oncogene addiction models of NSCLC, mainly in *EGFR*/*KRAS*/*ALK*-mutation-negative patients (Figure 1) [26]. Interestingly, the most frequent *HER2* mutations (90% of cases) are in-frame non-frameshift insertions in exon 20 of the tyrosine kinase domain (ex20ins), and some studies have described *HER2* mutations as the main factor in primary resistance to pan-HER TKIs, especially the YVMAins variant [51].

#### 2.3.2. HER2 Evaluation Criteria

Since no standardized evaluation methods have been developed, IHC/FISH guidelines for breast carcinomas are typically applied to lung cancer (Table 1).

#### 2.3.3. Prognostic Role of HER2 Alterations and Association with Clinicopathologic Features

The mutational profile of HER2 is associated with specific clinical features with prognostic value. In particular, primary *HER2* amplification is observed in 1–3% of NSCLCs and is associated with pleural metastases, whereas acquired *HER2* amplification is found in 10–15% of patients with acquired resistance to EGFR tyrosine kinase inhibitors (TKIs). HER2 protein overexpression is reported in 2–38% of NSCLCs and is associated with poor prognosis. *HER2* mutations occur in 1–4% of NSCLCs; they are predominant in women with no history of smoking and adenocarcinoma histology.

#### 2.3.4. Predictive Role of HER2 Alterations and Clinical Trials

Due to the plethora of studies concerning HER2 and its actionable therapy in NSCLC, only the published trials in the last 10 years enrolling more than 20 patients are reported with descriptions of the available drugs for *HER2* mutations, gene amplifications, and protein overexpression (Table 2) [26,27,28,29,30,31,32,33,34,35,36,37,38,39,40,41,42,43,44,45,46,47]. A total of 24 studies were identified, among which seven were retrospective studies and the others were phase II trials. Ten of them enrolled patients in the first-line treatment setting, and the others in later lines of treatment. Activity was the most frequent primary objective, but survival data were also reported.

Pan-HER TKI drugs against *HER2* mutation in pre-treated NSCLC patients were the first investigated strategy, with no significant improvement in terms of survival data [27,28,29,30]. Subsequently, selective HER2 TKIs were tested, with little improvement in terms of ORR and OS, but therapy-related adverse events (TRAEs) limited their approval [31,32,33,34,35,36].

Recently, ADCs have been approved by the Food and Drug Administration (FDA) for patients with advanced and metastatic pre-treated *HER2*-mutant NSCLC [51]. In the context of NSCLC, T-DM1 was the first ADC to be tested, and it showed the best ORR and PFS in comparison to other HER2 drugs and/or conventional therapy and anti-EGFR TKIs; however, although promising short-term efficacy, the response duration was limited [37,38,39]. Recently, T-DXd in recurrent or refractory NSCLC harboring *HER2* mutations or protein overexpression showed better results with an ORR of 55%, an mPFS of 8.2 months, a median duration of response (mDOR) of 9.3 months and a mOS of 17.8 months in the mutant cohort [40]. DESTINY-Lung02 confirmed T-DXd efficacy for pre-treated patients with *HER2*-mutant NSCLC and investigated appropriate dose regimens to reduce TRAEs such as interstitial pneumonia (5.9% vs. 14%), confirming that T-DXd may become a treatment of choice in this disease [41]. Moreover, a triple monoclonal therapy regimen (trastuzumab, pertuzumab, and docetaxel) was investigated in pre-treated HER2-mutant NSCLC, showing a better response duration compared with T-DXd [42]. Finally, due to low tumor mutation burden (TMB) in *HER2*-mutated NSCLC and subsequent low PD-L1 expression, the efficacy of immune checkpoint inhibitor (ICI) monotherapy in pre-treated patients and in first-line ones is limited and controversial [43,44].

Trastuzumab in *HER2*-amplified NSCLC has not demonstrated the same efficacy as it has in breast cancer [51]. Moreover, both T-DM1 in HER2-overexpressed (IHC 3+) NSCLC and T-DXd in *HER2*-amplified NSCLC have shown an ORR of 20% and 24.5%, respectively, which is lower for T-DXd in *HER2* mutant NSCLC [40,45]. Similar results were reported in *HER2*-amplified NSCLC receiving pyrotinib [32].

In conclusion, *HER2* mutations seem to have a significant predictive value for anti-HER2 therapy in NSCLC; for this reason, large biomarker screening programs, such as the French National Program or the US Lung Cancer Mutation Consortium (LCMC), have proposed systematic testing for *HER2* mutations in NSCLC [52].

### 2.4. Biliary Tract Cancer

#### 2.4.1. Epidemiology and Frequency of HER2 Alterations

Biliary tract cancers (BTCs) represent a rare and aggressive group of tumors arising from the bile ducts; in particular, GLOBOCAN data available for gallbladder cancer (GBC) report that it ranks 22nd for incidence and 20th for cancer-related mortality worldwide, respectively [7]. The term BTC includes cholangiocarcinoma (CCA), which can be subclassified in intrahepatic CCA (iCCA) and extrahepatic CCA (eCCA) based on its anatomic origin, GBC, and ampulla of Vater cancer (AVC). In recent years, the wide use of molecular sequencing techniques has revealed a great genomic heterogeneity in the landscape of BTCs, opening the possibility of a precision medicine approach with the use of specific target agents. The activation of HER2 can be observed in subsets of patients with BTCs. In particular, HER2 overexpression or gene amplification occurs in up to 20% of cases of GBC and eCCA, while its detection is low in iCCA. *HER2* mutations, on the other hand, are less frequent in BTCs, accounting for 1–2% of the total, while they reach a 7% rate in AVC [53] (Figure 1). Rates of HER2 overexpression seem to be different in Western and Eastern populations. In a Japanese study, 454 cases of BTCs were assessed for HER2 positivity, showing percentages ranging from 3% in iCCA to 31.3% in GBC [54]. Conversely, a large German retrospective study evaluated 436 samples for HER2 positivity and reported a total prevalence of 1.4% [55]. Lastly, a systematic review and meta-analysis specifically evaluating the role of the HER2/HER3 pathway in CCA described different prevalences of HER2 overexpression in Asian patients (28.4%) compared with Western ones (19.7%) [56].

#### 2.4.2. HER2 Evaluation Criteria

Molecular determination in BTCs mainly relies on next-generation sequencing (NGS) techniques for the possibility of simultaneously testing multiple gene alterations. However, HER2 assessment is more frequently analyzed using IHC staining, with criteria derived from other gastrointestinal cancers and no definitive consensus on its interpretation [57] (Table 1). Positivity, defined as either an IHC of 3+ or 2+ with *HER2* amplification detected using chromogenic in situ hybridization (CISH), was detected more frequently in distal bile duct tumors (2.4%).

#### 2.4.3. Prognostic Role of HER2 Alterations and Association with Clinicopathologic Features

The prognostic role of HER2 alterations in BTCs has been assessed in several studies with conflicting results. A retrospective Italian study included 100 resected BTC cases and showed a significantly shorter DFS in HER2-positive patients compared with HER2-negative tumors (10.6 vs. 20.9 months, *p* = 0.017), even though mOS was not statistically different between the two groups (*p* = 0.068) [58]. Similarly, in a retrospective Asian study, HER2 aberrations did not demonstrate statistical significance as an independent factor [59]. Moreover, HER2 status did not correlate with response to first-line cisplatin-gemcitabine (33.3% vs. 26.2%, *p* = 0.571) or survival (*p* = 0.776). No specific associations with clinic-pathological features have been described in the literature.

#### 2.4.4. Predictive Role of HER2 Alterations and Clinical Trials

Similarly to other solid cancers, HER2 overexpression has been extensively studied as a therapeutic target with the use of anti-HER2 agents. Single-agent lapatinib was tested in two phase II trials conducted in unselected BTC patients, with no results in terms of efficacy. Eight studies showing promising results in different settings and study phases are shown (Table 3) [24,60,61,62,63,64,65,66]. The multicenter phase II MyPathway basket trial evaluated the use of the double anti-HER2 blocked trastuzumab and pertuzumab in 39 patients with advanced BTC. ORR and disease control rate (DCR) was 23% and 51%, respectively, with an mDOR of 10.8 months and an mPFS of 4.0 months [60]. In another phase II basket trial evaluating the pan-HER TKI neratinib, the ORR was 12% in patients with BTC expressing *HER2* somatic mutation [61]. The use of trastuzumab in addition to chemotherapy has been tested in patients with BTCs who progressed on first-line cisplatin-gemcitabine. Among the 34 patients enrolled, 10 had a partial response (PR), and 17 had stable disease (SD) as the best response. ORR was 29.4%, mPFS was 5.1 months, and mOS was 10.7 months [62]. Similarly, the combination of tucatinib and trastuzumab showed clinically significant antitumor activity in a phase II basket trial [63]. Zanidatamab, a bispecific antibody that targets the two epitopes of HER2 bound by trastuzumab and pertuzumab, was tested in a phase I study, and subsequently, it was specifically assessed in BTC in the phase IIB HERIZON-BTC-01 trial [64,65]. The primary endpoint of ORR was 41.3% in the 80 patients enrolled in cohort 1. After a median follow-up of 12.4 months, mDOR was 12.9 months, DCR was 68.8%, and mPFS was 5.5 months. Lastly, T-DXd has been tested in patients with HER2-positive unresectable or recurrent BTC [66]. In patients with an IHC of 3+ and 2+, ORR was 36.4% (8/22; two CR and six PR). Encouraging signs of efficacy have also been observed in patients with HER2-low BTC, with an ORR of 12.5% (1/8; 1 PR) and a DCR of 75.0%. Median PFS and mOS were 4.2 months and 8.9 months, respectively. Updated results of the phase II multicohort DESTINY-PanTumor02 trial have been recently presented at the ASCO Annual Meeting [24]. In the BTC cohort, 41 patients received T-DXd, and 27 of those (65.9%) had received at least ≥2 prior treatment regimens. In the whole cohort, ORR was 26.8%, and mPFS was 4.1 months. Patients with an IHC of 3+ showed more favorable results with T-Dxd than patients with an IHC of 2+ (ORR 56.3%, mPFS 7.4 months vs. 4.2 months).

### 2.5. Colorectal Cancer

#### 2.5.1. Epidemiology and Frequency of HER2 Alterations

Colon cancer ranks 4th for incidence and 5th for cancer-related mortality worldwide, respectively, while rectal cancer ranks 8th for incidence and 10th for cancer-related mortality worldwide, respectively [7]. CRC shows a frequency of HER2 overexpression ranging between 3 and 5% (in the metastatic setting), while somatic *HER2* alterations, including amplifications (in about 5.8%), are found in 7% of patients, as identified in The Cancer Genome Atlas (TCGA) project (Figure 1) [77]. Mutations in *ERBB2* are found in 4–6% of patients with CRC, and some of these are activating and can co-occur with *HER2* amplification; however, unlike amplifications, *ERBB2* mutations are not associated with the *RAS*/*BRAF* wild-type status [78,79].

#### 2.5.2. HER2 Evaluation Criteria

HER2 testing in colorectal cancer (CRC) has proven to be fraught with difficulties in consensus, as different pathologists and laboratories utilize diverse protocols, none of which have become the accepted standard [69]. This lack of uniformity in interpretation means that clinical trials cannot be reliably compared and that the pathologist in the workplace may have uncertainties in deciding which evaluation protocol must be used. The lack of clear guidance on the optimal scoring criteria is a barrier to standardized and routine HER2 testing for CRC in clinical practice. This is mostly due, as already mentioned, to the availability of various assays or technical approaches, as underlined by the recent audit on HER2 testing in CRC performed in the US [69]. Indeed, as shown in a real-world setting, few laboratories are testing HER2 routinely in CRC, and of those that do, most (82.3%) use IHC as the primary test, with reflex to ISH for equivocal results. The main HER2 scoring systems that are being used include guidelines for HER2 testing in breast cancer and in gastric/gastroesophageal (GE) cancer (both ASCO/CAP in the US and ESMO guidelines in Europe) and the HER2 scoring system developed for CRC in the HERACLES clinical trial (Table 1). The ASCO/CAP breast cancer guidelines [2], though the first to be produced, have some advantages, such as equivalent evaluation between Ventana 4B5 and HercepTest antibodies and the fact that their application is widespread. However, only circumferential positivity is considered in breast cancer HER2 evaluation. While CRC often shows basolateral expression, a score of 3+ is often overcalled (as the positivity cut-off is 10% of cells), and, for these reasons, the use of these guidelines is not endorsed.

The use of HER2 evaluation guidelines developed in GE cancer for CRC, again, has similar advantages to breast cancer guidelines, as it can be performed by any approved IHC assay while also evaluating basolateral membrane HER2 expression. Similarly, though a 10% positivity cut-off is implemented [3]. HERACLES Diagnostic Criteria [4] were specifically developed for the HERACLES trial, a phase II trial testing the combination of trastuzumab and lapatinib in ERBB2-positive metastatic colorectal cancer patients refractory to standard treatment, including cetuximab or panitumumab [80]. The strengths of this approach were the fact that diagnostic criteria were shared between a Consensus Panel of Pathologists, they were established specifically for CRC and centrally reviewed, both routinely-used IHC clones were tested, and all test/validation cases were sent both to IHC and SISH/FISH [4]. Of note is that the Ventana 4B5 antibody was found to outperform HercepTest, especially with regard to sensitivity (HercepTest results in occasional false negatives) and concordance with ISH. According to these criteria, ERBB2-positive tumors showed ERBB2 immunostaining consisting of intense membranous ERBB2 protein expression, corresponding to homogenous *ERBB2* amplification, in >50% of cells. Of note, in this system, a score of 3+ expression of ≥10% but <50% neoplastic cells requires confirmatory ISH testing for amplification. Consequentially, disadvantages of this testing strategy include that these cases will require ISH testing while displaying a score of 3+, making a discussion with clinicians mandatory. Interestingly, Liu et al. showed that if a comparison is made between the HERACLES testing strategy and the GE criteria, a significant impact of HER2 positivity on survival is found only using HERACLES criteria, while no impact on survival is seen using GE criteria [81]. A further question that will need to be addressed in the future is the evaluation of HER2 biomarker status in “HER2-low” CRC. The study by Lang-Schwarz et al. [82] found only moderate interobserver agreement on HER2-low CRC by HERACLES criteria (much as has been seen in gastric cancer) [82,83]. The absence of a standardized reporting protocol for HER2 in CRC has impacted clinical trial data, as different trials have used different testing protocols, testing methods, and algorithms.

#### 2.5.3. Prognostic Role of HER2 Alterations and Association with Clinicopathologic Features

Patients with HER2-positive CRC have peculiar clinicopathologic features. Indeed, *HER2* amplification is more frequently found in CRC, with 65% to 90% of HER2-positive CRCs presenting in the left colon or rectum [84]. Moreover, *HER2*-amplified tumors often show a greater number of metastatic sites, most frequently lung and brain. The prognostic role of HER2 in CRC has been a controversial topic in the literature, with some studies showing reduced recurrence-free and OS for HER2-positive CRC, while others have not [4]. Results showed that patients whose tumors were HER2 wild-type tumors (*n* = 220) had significantly longer OS than patients (*n* = 13) with *HER2*-amplified CRCs (mOS 515 vs. 307 days, respectively; *p* = 0.0013)

#### 2.5.4. Predictive Role of HER2 Alterations and Clinical Trials

From a predictive point of view, apart from its well-known use as a predictive tissue biomarker for anti-HER2 treatment, and its amplification or overexpression, as well as *HER2*-activating mutations, have been associated with a lack of response to anti-EGFR drugs, as seen in a cohort of 233 patients with CRC receiving cetuximab [85]. Furthermore, acquired amplifications of *HER2* are associated with the development of resistance to EGFR-targeted therapies [86].

Also, in this context, HER2 expression has been studied as a therapeutic target for anti-HER2 agents. Two of the first anti-HER2 phase II trials involving patients with treatment-naive or pre-treated trials in CRC evaluated trastuzumab, in combination with chemotherapy, obtaining some degree of ORR; however, both were prematurely closed (the first due to low efficacy and the second due to low patient accrual) [67,68]. The already mentioned HERACLES-A trial was, however, successful in accrual; even though 914 patients were screened, 48 (5%) resulted in HER2-positive, and 27 were finally enrolled. The results from the HERACLES-A trial were encouraging, and new clinical trials have given favorable results in this setting, as summarized in Table 3 [67,68,70,71,72,73,74,75,76,80]. All of them are phase II trials, and only one included patients in first-line treatment settings, while the others enrolled patients in second or later lines of treatment. The HERACLES-B trial tried to assess the efficacy of the combination of pertuzumab and T-DM1, but it did not reach the primary endpoint of ORR [71]. However, ORR and SD ≥ 4 months were associated with higher HER2 IHC scores (3+ vs. 2+). In the MyPathway basket trial, results similar to those of the HERACLES-A trial were registered [70]. Subsequently, the promising activity of trastuzumab *plus* pertuzumab was confirmed in two other phase II trials: the TAPUR and the TRIMPH trials [72,73]. Recently, in this context, ADCs have also shown interesting results. The DESTINY-CRC01 trial investigated the activity and safety of T-DXd in patients with refractory HER2-positive *RAS*/*BRAF* wild-type mCRC [74]. Patients were assigned to three cohorts according to HER2 expression level: 53 patients in cohort A (IHC 3+ or IHC 2+ and FISH–positive), seven patients in cohort B (IHC 2+ and FISH–negative), and 18 patients in cohort C (IHC 1+). In cohort A, a confirmed objective response was reported in 24 patients after a median follow-up of 27.1 weeks. No responses were seen in cohorts B and C. Updated data confirmed high activity of T-DXd in cohort A (ORR, 45.3%; DCR: 83.0%; mPFS: 6.9 months; mOS: 15.5 months). Interestingly, the activity of T-DXd was not impaired in a subgroup of patients harboring *RAS* mutation-positive ctDNA. More recently, the MOUNTAINEER study evaluated trastuzumab in combination with tucatinib [75]. The interim analysis of the initial 26 patients enrolled demonstrated an ORR of 52.2%; high concordance between breast and gastric HER2 expression criteria was described, and this fact correlated with response to trastuzumab/tucatinib. Another interesting combination is trastuzumab *plus* pyrotinib [76]. The interim analysis of the HER2-FUSCC-G study, an ongoing open-label non-randomized phase IIa study in patients with HER2-positive gastrointestinal tumors, was recently presented. The ORR was 45.5% in the entire population, and it was 55.6% in patients with RAS wild-type tumors. The sustained HER2 blockade and the exploration of new potential drugs or combinations for patients who develop resistance to anti-HER2 therapy remain an ongoing issue, with numerous clinical trials currently in progress.

### 2.6. Bladder Cancer

#### 2.6.1. Epidemiology and Frequency of HER2 Alterations

Urothelial carcinoma (UC) is an aggressive neoplasm and a significant cause of mortality and morbidity globally. More than 90% of UCs arise in the bladder. Recent data reported bladder carcinoma as the sixth cancer for incidence and the 10th for mortality [7].

A recent systematic review investigated HER2 expression in early and advanced UC, and, in studies addressing advanced disease, an average of 13% of tumors were HER2-positive (range 6.7–37.5%), 17.5% were HER2-low (13.4–56.3%), and 39.5% were HER2-negative [87]. The heterogeneity observed across studies may be attributed to the lack of standardization in both the staining process and the result in interpretation, along with changes in the ASCO/CAP scoring guidelines over time. Amongst early UC, a single report described a rate of HER2-positive cases of 60%. However, recent investigations showed that *HER2* amplification seems to occur more frequently in advanced disease and that lymph node metastases displayed HER2 positivity more frequently than their primary counterpart [87,88,89]. *HER2* amplification was identified in 7.8% of advanced UCs (2.0–22.6%) and in 9% of early UCs, with a concordance between FISH and HER2 score of 3+ in 70% of UCs overall. Regarding *HER2* molecular alterations other than amplification, a single NGS report found that 16% of UCs in situ harbor missense mutations in the extracellular domain of *HER2*, among which the most common is the activating mutation S310F (Figure 1) [90]. In 11% of UCs overall, the mutation affects the Furin-like domain, a well-known mutational hotspot in the *HER2* gene [91]. HER2 enrichment in UC is more likely related to polysomy 17 rather than true amplification [89].

#### 2.6.2. HER2 Evaluation Criteria

No consensus has been reached for the evaluation of HER2 in UC, and most of the studies assessing HER2 expression relied on ASCO/CAP scoring guidelines for breast and gastric cancer (Table 1) [87]. Nevertheless, the HER2 expression pattern in UC has been described as a combination of gastric and breast cancer, with a prevalence of circular and patchy staining of tumoral cell membranes [88]. To date, the most common assays used were Dako HercepTest and Ventana 4B5 for IHC, Abbott PathVysion HER2 DNA Probe Kit, FoundationOne CDx, and Guardant360 CDx for FISH.

#### 2.6.3. Prognostic Role of HER2 Alterations and Association with Clinicopathologic Features

Disease-related factors associated with HER2 overexpression include the site of the primary tumor and the Consensus Molecular Classification subtype, with a higher incidence observed in the upper urinary tract and luminal cancers, respectively [87,89]. Studies on HER2 expression among unconventional subtypes of UC reported overexpression in 56% of the micropapillary variant; the frequency of HER2 expression drops in the plasmacytoid (25%) and squamous (20%) variants, and no staining was observed in the sarcomatoid variant [92]. In micropapillary UC, HER2 overexpression is only minimal (12%) due to gene amplification; in fact, an activating HER2 mutation was observed in 40% of cases.

In a recent meta-analysis investigating the relation between HER2 expression and clinicopathologic features of UC, a correlation with carcinoma in situ, multifocality, tumor size, grade, stage, lymph node metastases, progression, and recurrence was observed [93]. However, no significant difference in terms of OS or disease-specific survival was found between the two groups. A prospective study addressing survival outcomes in a cohort of 60 patients receiving standard surgical and medical treatment for muscle-invasive bladder UC reported a statistically significant difference between HER2-positive and HER2-negative tumors in terms of both OS and DFS, with HER2-positive UC having a worse prognosis [94]. Conflicting results were reported in a previous meta-analysis by Zhao et al., in which the role of HER2 positivity as a negative prognostic marker in UC was described in terms of dismal disease-specific survival and DFS [95]. Similarly, as highlighted in a recent review by Sanguedolce and colleagues, HER2 expression is also associated with increased recurrence rates and worse cancer-specific survival in early bladder UC [89].

#### 2.6.4. Predictive Role of HER2 Alterations and Clinical Trials

Several HER2-targeting agents have been tested in UC, mostly in an advanced setting (Table 4) [27,88,96,97,98,99,100,101,102,103,104,105,106,107]. We identified 15 clinical trials enrolling four or more UC patients. Among them, 2/15 were phase I, 1/15 phase I/II, 11/15 phase II, and 1/15 phase III studies. Five clinical trials were multi-basket studies, including different solid tumors. Administrated drugs were TKIs in 6/15 investigations, trastuzumab *plus* chemotherapy in 4/15, and ADCs in 5/15. In 13/15 studies, patients had already received prior lines of therapy, while only two trials enrolled patients who had received no prior therapy. IHC and FISH testing for HER2 was evaluated according to the ASCO/CAP guidelines for breast carcinoma. Primary endpoints included overall response, measured either by ORR or best overall response rate (7/15), treatment toxicity and safety (4/15), and PFS (4/15). A clear predictive role of HER2 expression in this context has not been demonstrated. However, although many clinical trials failed to demonstrate the significant clinical efficacy and safety of several HER2-targeted therapies in UC, ADCs have shown interesting preclinical and clinical results. Recently, in several ongoing trials, a newly developed ADC class combining HER2-targeted antibodies and immunotherapy is on the rise for the treatment of advanced UC, showing promising preliminary results [108].

### 2.7. Prostate Cancer

#### 2.7.1. Epidemiology and Frequency of HER2 Alterations

Prostate carcinoma (PC) is the fourth most frequent cancer and the eighth for mortality in the general population; in the male gender, current data report PC as the second cancer for incidence and the fifth for mortality [7]. Acinar adenocarcinoma represents the most common histotype, accounting for 95% of PCs. There is limited and conflicting evidence regarding HER2 expression in PC. HER2 positivity was reported in 8–18% of surgical specimens and 18% of transrectal ultrasound-guided biopsies (Figure 1) [117,118]. HER2 overexpression has been described in 10% of cases, but historical data have shown a difference in the frequency of HER2 positivity depending on disease stage and hormone resistance, with a stronger prevalence among metastatic, castration-resistant patients compared with localized, hormone-dependent PC patients (78% vs. 25%) [119]. Amplification has rarely been demonstrated using ISH techniques: one PC case resulted in amplified using FISH, and one case resulted in using CISH [120,121,122]. Only one previous study reported *HER2* amplification at a rate as high as 6% of PCs [109]. However, these data suggest the thesis that, in contrast with other types of tumors, HER2 expression is not related to *HER2* amplification in PC. Hence, HER2 expression in PC may be the result of other transcriptional processes, and the regulation of HER2 expression may have a post-transcriptional component. In particular, HER2 may provide an alternative mechanism for the activation of the androgen receptor signaling pathway through ligand-independent mechanisms in patients treated with androgen ablation therapy [109]. No mutation has been described for *HER2* in PC [121].

#### 2.7.2. HER2 Evaluation Criteria

In the vast majority of investigations, HER2 overexpression and amplification were assessed according to 2013 ASCO/CAP guidelines for breast cancer since specific criteria and reports in this context are lacking (Table 1). Among HER2-positive PC cases, only 2–6% of cases displayed strong membrane staining [120]. A large retrospective study conducted on prostatectomy specimens and testing HER2 positivity using two different monoclonal antibodies (Novacastra CB11 and Dako HercepTest) demonstrated HER2 positivity in only 1.5% of cases with both antibodies and a high concordance rate between the two tests was found [120]. In PC, the HER2 staining pattern was characterized by a focal reaction of isolated groups of cells with high tumoral heterogeneity. For this reason, Estephan and colleagues recently evaluated HER2 expression in PC through the HER2 scoring system used for gastric and gastroesophageal junction adenocarcinoma, reporting HER2 positivity in 23% of neoplasms [121].

#### 2.7.3. Prognostic Role of HER2 Alterations and Association with Clinicopathologic Features

HER2 expression showed a strong association with a high Gleason score (≥7), T stage, proliferative index Ki-67, and advanced disease [120]. Nevertheless, in other studies, no relation was found between HER2 expression and the aforementioned clinic-pathological features [117,122]. Furthermore, no association has been demonstrated between lymph node metastasis, serum prostate-specific antigen (PSA) level and surgical margins status [120]. HER2 overexpression was identified in a significant proportion of hormone-independent PC metastatic samples compared with androgen-dependent prostatic biopsy samples [123]. No clear prognostic role has been described for HER2 alterations in the context of PC.

#### 2.7.4. Predictive Role of HER2 Alterations and Clinical Trials

Few studies in the past decades have investigated the role of HER2-directed agents in HER-positive PC (Table 4) [110,111,123]. In all these clinical trials, HER2 status was evaluated using IHC and FISH according to the method validated for breast cancer. A phase II, prematurely closed trial failed to demonstrate clinical response in four PC patients who received trastuzumab *plus* docetaxel [110]. Another phase II trial reported two biochemical responses but no radiologic tumor expression among 18 patients with advanced hormone-independent PCs treated with trastuzumab alone [111]. A third phase II trial evaluated the difference in disease progression between a hormone-refractory PC treated with trastuzumab alone and a hormone-refractory PC treated with trastuzumab *plus* paclitaxel. Patients treated with trastuzumab showed disease progression, while about 50% of patients treated with a combination of trastuzumab *plus* paclitaxel experienced SD or a decline in PSA levels [123]. Considering the poor results of the clinical trials with HER2 agents in PC, preclinical trials with cancer models are now ongoing [124].

### 2.8. Gynecologic Cancer

#### 2.8.1. Epidemiology and Frequency of HER2 Alterations

Cervical Cancer (CeC) is the fourth most common cancer both in incidence and in mortality (6.8% and 8.1%, respectively) in women, and HPV is its main oncogenic driver [7]. On the other hand, Endometrial Carcinoma (EnC) is the 15th for incidence (2.1%) and mortality (1%) worldwide, whereas Ovarian Cancer (OC) is the 18th for incidence, with the highest mortality rate among gynecological cancers [7].

Though presenting a common gynecological origin, these malignancies differ in etiology, biology, and histological subtypes. In this context, a different *HER2* molecular profile has also been described, with the highest rate of *HER2* amplifications (>50% vs. 7% in OC vs. 5.14% in CeC) and HER2 overexpression (about 80% vs. 27% in OC vs. 21% in CeC) in EnC [6,125]. In particular, high-grade EnC, mostly represented by serous histotype, showed *HER2* amplification and HER2 overexpression in 10% to 40% of cases [126]. *HER2* mutations are very rarely described, with the highest rate in CeC (3% vs. 2% in EnC vs. 1% in OC) (Figure 1) [6].

#### 2.8.2. HER2 Evaluation Criteria

Similar to CRC, HER2 biomarker evaluation in gynecological cancer is less standardized and more controversial than in breast and gastroesophageal tumors. In the past, different criteria were used, but in March 2023, while not providing formal recommendations, the CAP suggested utilizing Buza et al.’s USC-specific criteria in their updated Template for Reporting Results of Biomarker Testing of Specimens From Patients With Carcinoma of Gynecologic Origin (Table 1) in the absence of more conclusive data [127].

#### 2.8.3. Prognostic Role of HER2 Alterations and Association with Clinicopathologic Features

The clinicopathologic features and prognostic significance of HER2 status in the molecularly classified PORTEC-3 trial population of patients with high-risk EnC were recently investigated [128]. HER2 positivity was documented in 25 cases (5.9%) out of 407. Interestingly, HER2 positivity was highly associated with the *p53*-abnormal subgroup (*p* > 0.0001), and this correlation was significantly stronger than between serous histology and the HER2 status (*p* > 0.0001), suggesting a molecular classification HER2-based is better than the morphologic one [128]. However, in multivariate analysis, including the molecular TCGA classification, *HER2* status did not have independent prognostic value.

Lack of well-defined guidelines for reporting *HER2* amplifications and overexpression, combined with a high heterogeneity of *HER2* amplification and overexpression, both at the intra-neoplastic level and in different histotypes, leads to a weak understanding of its prognostic value in gynecologic tract tumors.

#### 2.8.4. Predictive Role of HER2 Alterations and Clinical Trials

Conflicting results are described in HER2 target therapy trials without a clear role of HER alterations as predictive biomarkers (Table 4) [24,102,112,113,114,115,116]. The first studies documented no clinical benefit in patients treated with HER2 target therapy in EnC [102,112] and in OC [102,113]. On the other hand, patients who received trastuzumab and carboplatin/paclitaxel showed a better PFS with respect to carboplatin/paclitaxel alone, leading to the recommendation in the NCCN guidelines from carboplatin, paclitaxel, and trastuzumab combination therapy (Category 2A for advanced or recurrent EnC, sierous histotype) [114,129]. Finally, the most recent study documented a clinical benefit of pertuzumab *plus* trastuzumab in EnC and of T-DxT in EnC, CeC, and OC, with a good safety profile [24,115,116].

### 2.9. Other Malignancies

#### 2.9.1. Thyroid Cancer

Thyroid carcinoma (TC) is the seventh malignant tumor for incidence and the 24th cancer-related cause of death worldwide [7]. A wide variation in HER2 overexpression was reported in studies performed in thyroid cancer cells and tissues, with positivity rates varying from 0% up to 70%, which may largely be attributed to inter-study technical and interpretive variations. No data regarding *HER2* mutations are present in the literature. There are no standardized and validated criteria for scoring HER2 in this context, and due to these conflicting findings, there is no consensus regarding the potential prognostic and therapeutic value of this marker in TC in the currently available literature [130]. More recently, HER2 expression has been linked to the expression of estrogen receptors in thyroid tumor tissue and associated with *BRAF* V600E mutation and a more aggressive phenotype in familial papillary thyroid carcinoma (PTC). A study by Ruggeri et al. showed HER2 amplification/overexpression in 44% of follicular TCs and 18% of PTCs; moreover, HER2 overexpression was associated with a predictive factor in differentiated TCs (both papillary and follicular) with a tendency toward distant metastasis [131]. Another study by Syrai AK et al. demonstrated that HER2 overexpression occurs at a relevant frequency in PTC and in the absence of gene amplification [132]. Decreased HER2 expression was described in anaplastic and poorly differentiated TC correlating with aggressive behavior [133]. Few trials have investigated the activity of anti-HER2 agents in TC. A phase I trial assessed lapatinib in combination with dabrafenib for patients with advanced TC [134]. The investigators included 15 patients with *BRAF* V600 mutations (13 with well-differentiated TC and two with anaplastic TC). The reported PR was 60%, with an mPFS of 15 months. In an ongoing trial, neratinib (NCT 03065387) is being assessed for its effect on advanced solid tumors harboring any *HER* mutation, including TC [135].

#### 2.9.2. Renal Cell Carcinoma

Renal cell carcinomas (RCCs) comprise a large variety of neoplasms with substantial differences in morphology, pathogenesis, molecular alterations, and prognosis. As a whole, RCC represents the 14th and 16th cancer for incidence and mortality worldwide, respectively. [7] At present, there are conflicting reports concerning HER2 status in RCC. In particular, the frequency of HER2 overexpression throughout different investigations ranged from 11% to 64% of RCCs [136]. These discrepancies in rate may be partly due to differences in staining techniques adopted for IHC evaluation and cohort compositions. Only a single investigation reported HER2 amplification using PCR in 17% of cases, but this finding was not supported further [137]. Polysomy of chromosome 17, where *HER2* is located, with concurrent increased *HER2* copy number, was described in 26% of RCCs [138]. No mutation in the *HER2* gene has been reported. Also, in this context, no standardized and validated criteria for scoring HER2 are present in the literature. Some studies described that the highest frequency of IHC HER2 expression was found in chromophobe and papillary subtypes in some studies and in clear cell RCC in others [137,139]. Of note, Selli et al. reported *HER2* amplification at competitive PCR in 45% of collecting duct carcinomas, a very aggressive, albeit rare, subtype of RCC [140]. It is unclear whether HER2 may have a role in RCC pathogenesis or if its eventual expression may be only an epiphenomenon of other pathways’ activation. When compared to adjacent non-neoplastic parenchyma, HER2 staining in the tumor resulted in reduced/absence. In particular, score 3+ positivity was confined to the distal part of the nephron, while proximal tubules, which are supposed to give rise to the major part of RCCs, were negative [138]. Consistent with this, an inverse correlation between tumor grade and HER2 immunostaining was described, suggesting that the reduction in HER2 expression may play a role in RCC oncogenesis [141]. In contrast with this finding, other investigations have demonstrated higher levels of HER2 mRNA and more frequent HER2 positivity at IHC in high-grade RCCs [137]. Lastly, no association between HER2 expression and tumor grade and stage has been reported [139]. Owing to these conflicting reports, HER2 expression has rarely been considered a potential biomarker for targeted therapy in this context, and, to date, only one prospective, randomized, phase III, open-label trial has been performed [142]. This study compared lapatinib with hormone therapy in patients diagnosed with locally advanced or metastatic RCC. Tumors were analyzed with IHC for EGFR and HER2 expression, using breast cancer guidelines for the evaluation of HER2. However, this study showed the equivalent efficacy of lapatinib and hormone therapy, with a prolonged OS limited to an EGFR score of 3+ RCCs.

#### 2.9.3. Pancreatic Ductal Adenocarcinoma

Pancreatic cancer ranks 12th for incidence and sixth for cancer-related mortality worldwide [7]. The amplification of the *HER2* gene and/or the overexpression of HER2 protein have been implicated in the development of pancreatic ductal adenocarcinoma (PDAC). According to the literature, the prevalence of HER2 overexpression in patients with PDAC ranges from 4 to 50% [143,144]. By contrast, only a minority of cases have shown *HER2* mutation or amplification, indicating a conceivable discrepancy between genetic alteration and protein expression of HER2 in PDAC [145]. In 90% of pancreatic cancers, HER2 protein overexpression is attributable to gene amplification. No data regarding HER2 mutations are present in the literature. For BTCs, the criteria used for scoring HER2 using IHC in PDAC were the same as those used for gastric cancer, but this technique could lead to false-positive cases and failed as a prognostic biomarker (Table 1). Similarly, *HER2* gene amplification does not represent a good prognostic factor for survival in patients with PDAC, according to different meta-analyses [146,147]. A retrospective study conducted on 55 patients with PDAC showed 7.3% of HER2 IHC 3+, with nine cases originally classified as negative for HER2 using IHC based on the gastroesophageal IHC scoring scheme presenting *HER2* gene amplification. In this study, patients without HER2 expression showed better prognosis compared with HER2 overexpressed tumors (*p* = 0.027) [148]. Regarding the efficacy of anti-HER2 agents in metastatic PDAC results from studies evaluating the addition of trastuzumab with capecitabine or gemcitabine did not show any improvement in terms of OS [130,149]. Similarly, disappointing results have recently come from the DESTINY-PanTumor02 trial [24]. Twenty-five patients have been treated with T-DXd, with an ORR of 4% in the whole population (0% in IHC 3+ cases and 5.3% in IHC 2+ ones). Median PFS was 3.2 months (1.4–7.0), rising to 5.4 months (2.8–NE) in two patients with IHC 3+.

#### 2.9.4. Hepatocellular Carcinoma

Primary liver cancer ranks sixth for incidence and third for cancer-related mortality worldwide, respectively, and hepatocellular carcinoma (HCC) accounts for 75–85% of cases [7]. HER2 expression has been reported to be between 0% and 29.6% [150]. No data regarding *HER2* mutations exist, and no validated criteria for HER2 testing are present. Some studies did not identify any association between HER2 expression and clinicopathologic features nor any relation to tumor development [151]. A recent study showed that HER2 expression is a rare event (7.4%) in HCC, and no relation between clinicopathologic features and HER2 expression was established [150]. Despite that, since HER2-targeted therapy is evolving, it is important to continue this investigation, and three clinical trials with anti-HER2-targeted therapies are now open [152].

#### 2.9.5. Small Bowel Adenocarcinoma

Small bowel cancers are rare and represent less than 5% of gastrointestinal cancers [153]. The four most common histological types of cancer in the small bowel include adenocarcinomas, neuroendocrine tumors, gastrointestinal stromal tumors, and lymphomas. Small bowel adenocarcinoma (SBA) constitutes around 40% of small bowel cancers and is associated with a grim prognosis, with the 5-year OS ranging from 26 to 40%. The duodenum is the most commonly affected segment, representing 55–82% of cases, followed by the jejunum and the ileum. The frequency of *HER2* alterations in SBA varies widely in the literature, being found in up to 23% of cases in a study [154]. Importantly, *HER2* point mutations represent the most common *HER2* alterations, occurring in 7–14% of SBAs [155,156], whereas amplifications are rarer, occurring in about 4% of SBAs and accounting for only a quarter of *HER2* alterations in SBAs, differently from colorectal cancer [154,155,157,158]. In the few studies where HER2 expression evaluation was performed, breast or gastric cancer criteria were applied for both IHC and ISH scoring (Table 1 and Figure 2) [155,158]. *HER2* mutations have been consistently associated with mismatch repair deficiency in SBAs, and, in some but not all studies, they are more frequent in duodenal cancers [154,155]. SBAs associated with Crohn’s disease generally do not harbor *HER2* mutations, whereas they have been reported to have *HER2* amplification in up to 8% of cases [156,158]. Although no significant prognostic impact of *HER2* amplification or mutation has been yet identified in SBA patients, in the work by Adam et al., patients with alterations in the *HER2* signaling cascade (64%) demonstrated worse clinical outcomes (mOS 70.3 vs. 109 months). Very few therapeutic experiences with anti-HER2 agents (three case reports) have been described in the literature, with potential clinical benefit [159,160,161]. A case report about the use of anti-HER2 targeted therapy in a HER2-positive patient with stage III duodenal SBA has been described [159]. After surgical resection, the patient started adjuvant chemotherapy and trastuzumab until completing 1 year. In another HER2-positive duodenal SBA patient with liver metastasis, trastuzumab, in combination with chemotherapy, was used before surgery as neoadjuvant therapy [160]. Since a complete instrumental response was evident, the patient underwent a classic pancreaticoduodenectomy with resection of segment 4B. Final pathologic analysis revealed no residual invasive adenocarcinoma, consistent with a complete neoadjuvant treatment response. A recent case report describes a 50-year-old female patient who was diagnosed with stage IV duodenal SBA with metastasis to both lungs and retroperitoneal lymph nodes [161]. The NGS revealed *HER2* amplification. Trastuzumab *plus* oxaliplatin-based chemotherapy was started as first-line treatment. The patient achieved PR and had a PFS of 6 months. After PD, the patient started the second-line treatment with trastuzumab and PD1 inhibitors and remained stable with a PFS of 3 months.

#### 2.9.6. Anal Cancer

Anal cancer ranks 30th for incidence and 30th for cancer-related mortality worldwide [7]. The first and only molecular analysis to identify HER2 expression and *ERBB2* mutations in squamous cell carcinoma of the anal canal has been recently published [162]. HER2 IHC was positive in 0.9% of cases, amplification using chromogenic ISH was seen in 1.3% of cases, and mutations in *ERBB2* were present in 1.8% of cases, with a potential for targeted therapy. No validated criteria for HER2 testing exist for anal cancer, and a predictive and prognostic role of HER2 alteration is not known, as well as a relationship with clinicopathologic features. No specific clinical trials have been performed yet.

#### 2.9.7. Non-Melanoma Skin Cancers

Non-melanoma skin cancer ranks fifth for incidence and 22nd for cancer-related mortality worldwide, respectively [7]. The role of HER2 in the carcinogenesis of basal cell carcinomas (BCCs) and squamous cell carcinomas (SCCs) is unclear. A study by Krahn et al. described that HER2 is ubiquitously expressed [163]. Isolated HER2 expression and EGFR/HER2 have predominantly been found in normal skin, while HER2/HER3 and the triple expression of EGFR/HER2/HER3 have more frequently been seen in BCCs and SCCs compared with normal skin (50% and 40% vs. 26%, respectively). No criteria for HER2 testing exist for these tumors, and the predictive and prognostic role of HER2 alteration is not known. No specific clinical trials have been performed yet. The activation of HER3, in addition to EGFR and HER2, might, therefore, be associated with the malignant phenotype. HER2 was also found to be present in about 61% of cases of extramammary Paget disease (EMPD), with a significant correlation between the presence of invasion and strong positivity (3+) for HER2 [164]. The results of the recent DESTINY-PanTumor02 trial included patients with EMPD with good ORR in the entire cohort of patients and, in particular, in the HER2+ IHC 3+ cases [24]. Another recent paper summarized the clinical course of 17 patients with HER2-positive advanced EMPD treated with anti-HER2 agents [165]. Results showed four cases of CR (20%) and nine of PR (45%). Most of the patients were treated with anti-HER2 therapy at the first line (12, 60%) in combination with taxane (eight, 66.7%), and they obtained a mPFS of 12 months. Also, a treatment with pyrotinib has been reported, resulting in a PR but with only 2 months of follow-up. No specific trials are ongoing in this context, but anti-HER2 therapies should be employed.

## 3. Conclusions

HER2-positive carcinomas themselves are generally characterized by an aggressive clinical course, but in some cases, patients benefit from HER2-targeting therapies with improvements in outcomes. Our study showed that HER2 testing is still investigational in several non-breast and non-gastroesophageal carcinoma types. The most advanced predictive pieces of evidence come from CRC, BTC, lung cancer, and SGC, and, therefore, HER2 testing in such cancers is recommended in the metastatic setting. However, not all patients with solid tumors showing either HER2 positivity (with IHC and/or gene amplification) or *HER2* mutation benefit from these treatments. Thus, more selection criteria of patient candidates to receive HER2-targeted therapies are required. It is important to identify which HER2 alteration may be targeted because *HER2* mutations and amplifications or protein expressions have revealed different prognostic and predictive roles in different organs. Major problems in the evaluation of HER2 positivity outside of breast and gastric cancer are (1) the lack of standardized and site-specific scoring systems and (2) *HER2* amplification is often assessed using different methods that are neither standardized nor comparable across different laboratories/trials. At present, HER2 positivity cannot be considered an agnostic predictive biomarker. The new frontier of the predictive role of “HER2-low” IHC expression, as well as the opportunity to repeat biopsy sampling at disease progression to re-assess HER2 status, deserve further investigation outside of breast and gastric carcinomas.

## Figures and Tables

**Figure 1 cancers-16-03145-f001:**
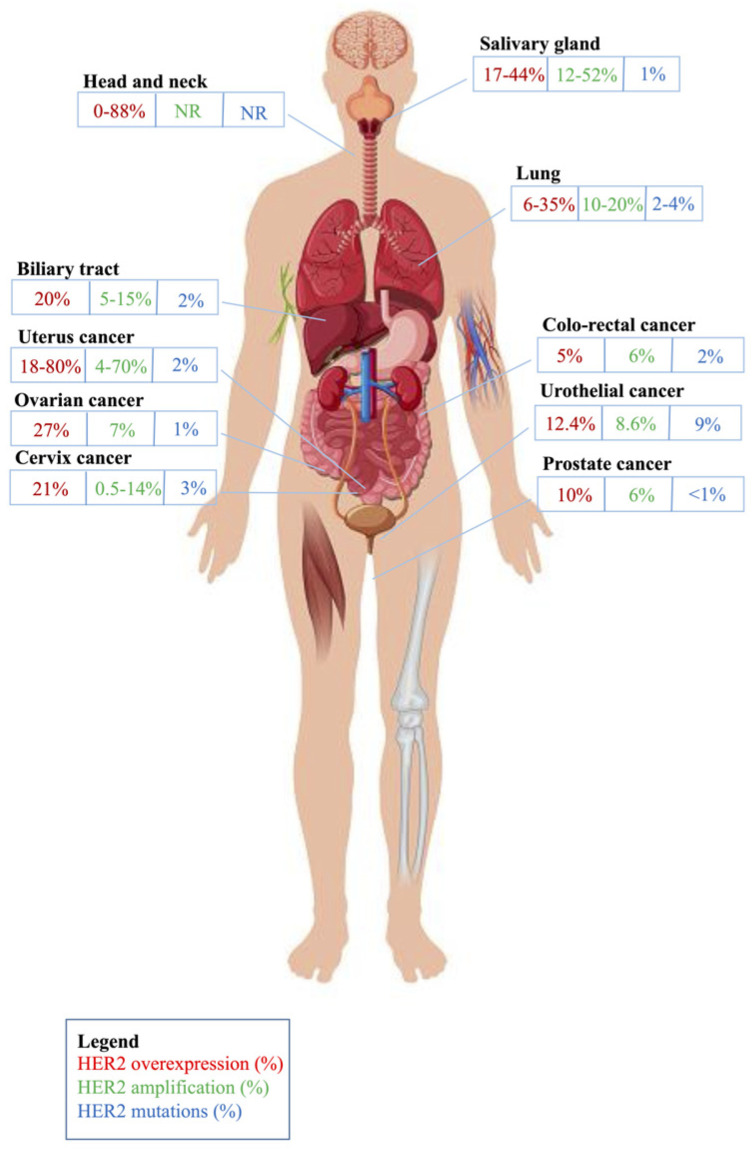
Frequency of HER2 alterations across the tumor types described in this review. Figure adapted from Matthew Cole/Vettoriale stock Alamy, 2016, with copyright permission.

**Figure 2 cancers-16-03145-f002:**
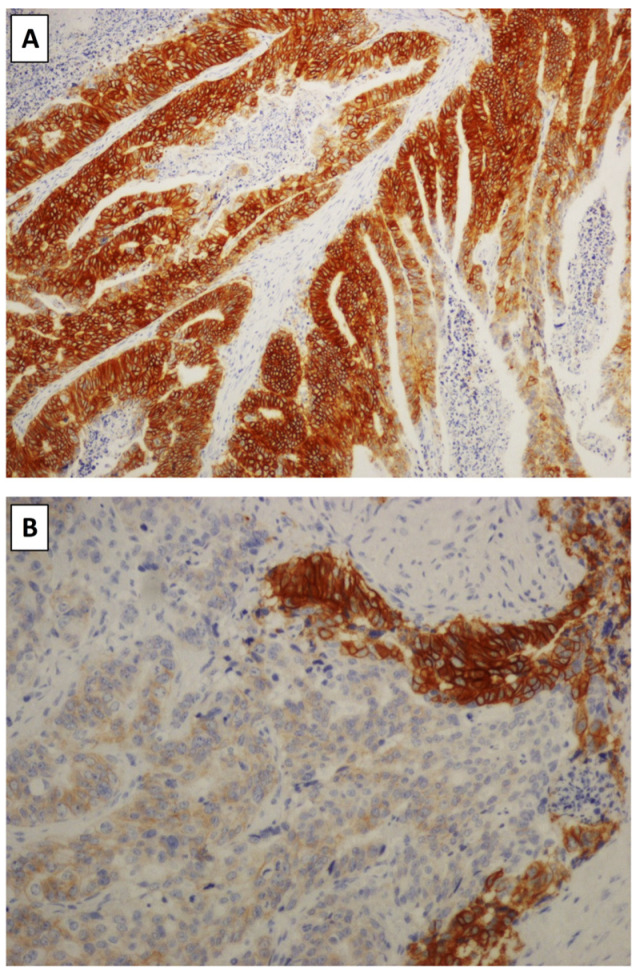
(**A**) Histology of a jejunal adenocarcinoma with a HER2 overexpression (IHC 3+). Note the strong membranous expression of HER2 in most tumor cells. (**B**) The same case of (**A**) shows a minor component of the score IHC 1+. Note that in the upper-right corner, there is an area of score 3+ expression.

**Table 1 cancers-16-03145-t001:** Immunohistochemical scoring systems and ISH definitions of HER2 evaluation across the different tumor types analyzed.

Type of Tumor	Recommended HER2 IHC Scoring Systems	Interpretation of HER2 IHC	Interpretation of ISH
Salivary gland carcinoma	Breast cancer criteria with Hercept test: **score 0:** no staining observed or Incomplete membrane staining that is faint or barely perceptible and within ≤10% of the invasive tumor cells **score 1+:** Incomplete membrane staining that is faint or barely perceptible and within >10% of the invasive tumor cells **score 2+:** Weak to moderate complete membrane staining observed in >10% of tumor cells **score 3+:** Circumferential membrane staining that is complete, intense, and in >10% of tumor cells	**Score 0 and 1+:** negative **Score 2+:** equivocal, need ISH confirmation **Score 3+:** positive	Breast cancer criteria: **ISH negative:** - *HER2*/CEP17 ratio < 2.0 with an average *HER2* copy number < 4.0 (group 5); - *HER2*/CEP17 ratio ≥ 2.0 and average *HER2* copy number < 4.0 (group 2) with concurrent IHC 2+; - *HER2*/CEP17 ratio < 2.0 with an average *HER2* copy number ≥ 4.0 and < 6.0 (group 4) with concurrent IHC 2+; - Groups 2, 3, and 4 with concurrent IHC 0 or 1+. **ISH positive:** - *HER2*/CEP17 ratio ≥ 2.0 and average *HER2* copy number ≥ 4.0 (group 1); - *HER2*/CEP17 ratio ≥ 2.0 and average *HER2* copy number < 4.0 (group 2) with concurrent IHC 3+; - *HER2*/CEP17 ratio < 2.0 and average *HER2* copy number ≥ 6.0 (group 3) with concurrent IHC 2+; - *HER2*/CEP17 ratio < 2.0 and average *HER2* copy number ≥ 6.0 (group 3) with concurrent IHC 3+; - *HER2*/CEP17 ratio < 2.0 with an average *HER2* copy number ≥ 4.0 and < 6.0 (group 4) with concurrent IHC 3+.
HNSCC	See salivary gland carcinoma	See salivary gland carcinoma	See salivary gland carcinoma
Lung cancer	See salivary gland carcinoma	See salivary gland carcinoma	See salivary gland carcinoma
Biliary tract cancer	Gastroesophageal cancer criteria with Hercept test: **Score 0:** No reactivity or membranous reactivity in <10% of tumor cells **Score 1+:** Faint or barely perceptible membranous reactivity in ≥10% of tumor cells; cells are reactive only in part of their membrane. **Score 2+:** Weak to moderate complete, basolateral or lateral membranous reactivity ≥ 10% of tumor cells **Score 3+:** strong complete, basolateral or lateral membranous staining in ≥10% of tumor cells	**Score 0 and 1+:** negative **Score 2+:** equivocal, ISH confirmation is needed **Score 3+:** positive	Gastroesophageal cancer criteria: **ISH positive:** - *HER2*: centromeric probe for chromosome 17 (CEP17) ratio of ≥2)
Colorectal cancer	CAP/ASCP/ASCO GEA: **Score 0:** no reactivity or membranous reactivity in <10% **Score 1+:** faint/barely perceptible reactivity in ≥10% **Score 2+:** weak to moderate complete, basolateral, or lateral membranous reactivity in ≥10% but <50% **Score 2+:** weal to moderate complete, basolateral, or lateral membranous reactivity in ≥50% **Score 3+:** strong complete, basolateral, or lateral membrane staining in 10–50% **Score 3+:** strong complete, basolateral, or lateral membrane staining > 50% HERACLES Diagnostic Criteria with Ventana 4B5: **Score 0:** no staining **Score 1+:** faint staining, any cellularity, segmental or granular pattern **Negative (2+):** moderate staining in <50% cells, any pattern **Equivocal (2+):** moderate staining in ≥50% cells, any pattern with circumferential, basolateral or lateral pattern (IHC mandatory: re-test IHC; if confirmed, proceed with ISH) **Negative (3+):** intense in ≤10% cells, circumferential, basolateral or lateral pattern **Positive (3+):** intense in >10% and <50% cells, circumferential, basolateral or lateral pattern (IHC mandatory: re-test IHC; if confirmed, proceed with ISH) **Positive (3+):** intense in ≥50% cells, circumferential, basolateral, or lateral pattern	CAP/ASCP/ASCO GEA: **Score 0 and 1+:** negative **Score 2+:** equivocal, need ISH confirmation HERACLES Diagnostic Criteria with Ventana 4B5: **Score 0 and 1+:** negative **Negative (2+):** moderate staining in <50% cells, any pattern **Equivocal (2+):** moderate staining in ≥50% cells, any pattern with circumferential, basolateral or lateral pattern (IHC mandatory: re-test IHC; if confirmed, proceed with ISH) **Negative (3+):** intense in ≤10% cells, circumferential, basolateral or lateral pattern **Positive (3+):** intense in >10% and < 50% cells, circumferential, basolateral or lateral pattern (IHC mandatory: re-test IHC; if confirmed, proceed with ISH) **Positive (3+):** intense in ≥50% cells, circumferential, basolateral or lateral pattern	CAP/ASCP/ASCO GEA and HERACLES Diagnostic Criteria: - *HER2*/CEP17 ratio ≥ 2 in ≥50% of cells
Urothelial cancer	See salivary gland carcinoma and biliary tract cancer	See salivary gland carcinoma and biliary tract cancer	See salivary gland carcinoma and biliary tract cancer
Prostate cancer	See salivary gland carcinoma	See salivary gland carcinoma	See salivary gland carcinoma
Gynecological cancer	**Score 2+:** intense complete or basolateral/lateral membrane staining in 30% or fewer tumor cells or weak to moderate staining in greater than or equal to 10% of tumor cells **Score 3+:** Intense complete or basolateral/lateral membrane staining in over 30% of tumor cells	**Score 2+:** equivocal, must order reflex test (same specimen using ISH) or order a new test (new specimen if available, using IHC or ISH). **Score 3+:** positive	**ISH negative:** - FISH *HER2*/CEP17 ratio < 2.0 and average *HER2* copy number less than 6 per nucleus **ISH positive:** - FISH *HER2*/CEP17 ratio ≥ 2.0; - FISH *HER2*/CEP17 ratio < 2.0 with average HER2 copy number equal to or greater than 6 per nucleus.
Thyroid cancer	See salivary gland carcinoma	See salivary gland carcinoma	See salivary gland carcinoma
RCC	See salivary gland carcinoma	See salivary gland carcinoma	See salivary gland carcinoma
PDAC	See biliary tract cancer	See biliary tract cancer	See biliary tract cancer
HCC	See biliary tract cancer	See biliary tract cancer	See biliary tract cancer
Small bowel adenocarcinoma	See biliary tract cancer	See biliary tract cancer	See biliary tract cancer
Anal cancer	See biliary tract cancer	See biliary tract cancer	See biliary tract cancer
Non-melanoma skin cancers	NR	NR	NR

**Abbreviations:** CEP17, chromosome enumeration probe 17; HER2, human epidermal growth factor receptor 2; HNSCC, head and neck squamous cell carcinoma; IHC, immunohistochemical; ISH, in situ hybridization; NR: not reported in the literature; PDAC, pancreatic ductal adenocarcinoma; RCC, renal cell carcinoma.

**Table 2 cancers-16-03145-t002:** Published papers with anti-HER2 agents in salivary gland carcinoma, HNSCC, and lung cancer.

Type of Tumor	Author, Year [Ref]	Study Design	N° pts	Treatment Line	Type of HER2 Alteration Evaluated	Definition of HER2 Positivity	Drug	Primary Endpoint	Results	Survival Data
Salivary gland carcinoma	Haddad R et al., 2003 [12]	Phase I	14	First-line	Overexpression	IHC 2+ or 3+ (breast criteria)	Trastuzumab	PFS	4.2 mo	See primary endpoint
	Locati LD et al., 2005 [13]	Retrospective	4	First- and second-line	Amplification/ Overexpression	IHC 3+ or 2+ confirmed by FISH (breast criteria)	Trastuzumab	Activity	SD 25%	mPFS: 2.5 mo
	Limaye SA et al. 2013 [14]	Retrospective	5	Adjuvant and first-line	Amplification/ Overexpression	3+ (strong complete membrane immunoreactivity in >30% of tumor cells) or 2+ (weak to moderate complete membrane immunoreactivity in at least 10% of tumor cells) with a FISH ratio > 2.2	Paclitaxel, carboplatin, *plus* trastuzumab	Activity	PR (2), CR (1), PD (2)	mOS: 40 mo
	Perissinotti AJ et al., 2013 [15]	Retrospective	13	Adjuvant and progressed on previous treatments	Amplification/ Overexpression	IHC 3+ or 2+ (breast criteria) with FISH ratio > 2.0 or an average number of *HER2* gene copies/cell of 6 or greater.	Trastuzumab or trastuzumab *plus* CT	Activity	no response to single-agent; PR (3) with combined treatments	NR
	De Block K et al. 2016 [16]	Retrospective	6	Progressed on previous treatments	Amplification/ Overexpression	IHC 3+ or 2+ confirmed by FISH (breast criteria)	Trastuzumab *plus* taxane	Activity	PR (5), SD (1)	mPFS: 10.8 mo
	Takahashi H et al., 2019 [17]	Phase II	57	Progressed on previous treatments (no antiHER2 agents)	Overexpression/ Amplification	IHC 3+ or gene amplification confirmed by FISH, according to the ASCO/CAP guidelines for breast cancer	Doce *plus* trastuzumab	ORR	70.2%	mPFS: 8.9 mo (95% CI, 7.8 to 9.9 months) mOS: 39.7 mo (95% CI, not reached)
	Kurzrock R et al., 2019 [18]	Phase II	15	Progressed on previous treatments (also anti-HER2 agents)	Amplification/ overexpression and mutation	IHC 3+ according to Breast cancer criteria 2013; or gene amplification with a *HER2*/CEP17 ratio of >2.0 or *HER2* gene copy number > 6.0 by ISH; or *HER2* gene copy number gain by NGS or RT-PCR.	Pertuzumab *plus* trastuzumab	ORR	63%	mPFS: 8.6 mo mOS: 20.4 mo
	Jhaveri KL et al., 2019 [19]	Phase II	3 SGC: 2 MCPG and 1 SCCPG	Progressed on previous treatments (no antiHER2 agents)	Amplification	*ErbB2* gene copy number > 7 by NGS	T-DM1	ORR	PR 5.6%: 1 MCPG and 1 SCCPG; SD 47%	6 mo PFS rate: 23.6% [90% CI 14.2% to 39.2%].
	Kawakita et al., 2022 [20]	Retrospective	111	Progressed on previous treatments (no antiHER2 agents)	Overexpression/ amplification	IHC 3+ or gene amplification by FISH according to the ASCO/CAP guidelines for breast cancer.	Doce *plus* trastuzumab	ORR	72%	mPFS: 9 mo (8–11 months); OS: 38 mo (33–49 months)
	Sousa LG et al., 2022 [21]	Retrospective	17	First and subsequent line of therapy	Overexpression/ amplification	IHC 3+ or 2+; FISH-positivity based on the breast cancer criteria (ratio > 2.2 or copy number > 6)	Trastuzumab *plus* CT	ORR	47%	mPFS: 9.6 mo (95% CI, 4.9–11.6%)
	Uijen MJM et al., 2022 [22]	Retrospective	13	First and second line	Overexpression/ amplification	IHC 3+ (strong expression in >10%) or 2+ with FISH ratio > 2 (breast criteria)	Doce, trastuzumab, and pertuzumab (1st line); T-DM1 (2nd line)	ORR	1st line: 58% 2nd line: 57%	1st line: mPFS: 6.9 mo (95% CI 5.3–8.5); mOS: 42.0 mo (95% CI 13.8–70.1). 2nd line: mPFS of 4.4 mo (95% CI 0–18.8).
	Lee J et al., 2022 [23]	Phase II	43	Progressed on previous treatments	Overexpression/ amplification	IHC 3+ (strong expression in >10%) or 2+ with FISH ratio > 2 (breast criteria)	Doce *plus* trastuzumab	ORR	69.8%	mPFS: 7.9 mo (6.3–9.5) mOS: 23.3 (19.9–26.7)
	Meric-Bernstam F et al., 2023 [24]	Phase II	19	Progressed on previous treatments (also anti-HER2 agents)	Overexpression/ amplification	IHC ≥ 2+ using current ASCO/CAP guidelines for scoring HER2 in gastric cancer	T-DXd	ORR	42.1%	mPFS: 12.5 mo
HNSCC	Meric-Bernstam F et al., 2024 [24]	Phase II	4	Progressed on previous treatments (also anti-HER2 agents)	Overexpression/ amplification	IHC ≥ 2+ using current ASCO/CAP guidelines for scoring HER2 in gastric cancer	T-DXd	ORR	50%	NR
NSCLC	Mazières et al., 2013 [26]	Retrospective	65	First and subsequent line of therapy	Mutations	PCR	CT + Trastuzumab/afatinib/ lapatinib/masatinib	OS	RR 50% DCR 80%	PFS: 5.1 mo OS: 40 mo
	Hyman DM et al., 2018 [27]	Phase II	26	First-line or later line	Mutations	NGS	Neratinib	ORR	ORR 3.8%	mPFS: 5.5 mo
	Besse B et al., 2014 [28]	Phase II	27	Second-line or later-line	Mutations	NGS	Neratinib +/− temsirolimus	ORR	ORR Neratinib 0% vs. Neratinib + Temsirolimus 21%	mPFS: Neratinib 2.9 mo vs. Neratinib + temsirolimus 4.0 mo
	Kris MG et al., 2015 [29]	Phase II	26	Second line	Mutations	PCR	Dacomitinib	OS	OR 12%	PFS: 3 mo mOS: 9 mo
	Peters S et al., 2018 [30]	Retrospective	28	Second line	Mutations	PCR	Afatinib	Activity	TTF 2.9 mo ORR 19% DCR 69%	NR
	Zhou C et al., 2022 [31]	Phase II	60	Second line	Mutations	NGS	Pyrotinib	ORR	ORR 30%	mPFS: 6.9 mo mOS: 14.4 mo
	Song Z et al., 2022 [32]	Phase II	78	First-line and later line	Mutations	NGS	Pyrotinib	PFS at 6 months	ORR 19.2%, mDoR 9.9 mo	mPFS: 5.6 mo mOS: 10.5 mo
	Yang G et al., 2022 [33]	Phase II	31/33	Second-line and later line	Mutations	32 NGS; 1 PCR	Pyrotinib + apatinib	ORR	ORR 51.5% DCR 93.9% mDoR 6.0 mo	mPFS: 6.9 mo mOS: 14.8 mo
	Le X et al., 2022 [34]	Phase II	90	Second- line and later line	Mutations	NGS	Poziotinib	ORR	ORR 27.8% DCR 70% mDoR 5.1 mo	mPFS: 5.5 mo
	Elamin YY et al., 2022 [35]	Phase II	30	First-line and later line	Mutations	NGS	Poziotinib	ORR	ORR: 27% RR (8 weeks) 43% DCR 73%	NR
	Sun S et al., 2022 [36]	Phase II	70	First-line	Mutations	NGS	Poziotinib	ORR	ORR 41% DCR 73%, mDoR 5.7 mo	mPFS: 5.6 mo
	Mazières J et al., 2016 [37]	Retrospective	58/101	Second-line and later line	Mutations	PCR/NGS	T-DM1	Activity	ORR 50.9% DC 75.5%	PFS: 4.8 weeks OS: 13.3 weeks
	Li BT et al., 2020 [38]	Phase II	49	First-line and later line	Mutations and/or amplifications	NGS and FISH	T-DM1	ORR	ORR: Mut: 50% Ampl: 55% Mut + Ampl 50%; mDoR 4.4 mo	PFS: 5.0 mo
	Iwama E et al., 2021 [39]	Phase II	22	Second-line and later line	Mutations	NGS or PCR	T-DM1	ORR	ORR 38.1% mDoR 3.5 mo	mPFS: 2.8 mo
	Li BT et al., 2022 [40]	Phase II	91	Second-line	Mutations	NGS	T-DXd	ORR	ORR 55% mDoR 9.3 mo	mPFS: 8.2 mo mOS: 17.8 mo
	Li BT et al., 2022 [40]	Phase II	91	Second-line	Overexpression	FISH breast	T-DXd	ORR	ORR 24.5% mDoR 6.0 mo	mOS: 11.3 mo mPFS: 5.4 mo
	Goto K et al., 2023 [41]	Phase II	152	Second-line	Mutations	NGS	T-DXd	ORR	mDoR 16.8 mo vs. NE; DCR 93.1% vs. 92%; ORR 49.0% vs. 56.0%	NR
	Mazieres J et al., 2022 [42]	Phase II	45	Second-line	Mutations	NGS	Trastuzumab, pertuzumab, doce	ORR	mDoR 11.0 mo ORR 29%	mPFS: 6.8 mo
	Mazieres J et al., 2019 [43]	Retrospective	29/551	First-line and later line	Mutations	NGS/Other	ICI	Activity	ORR 7%	mPFS: 2.5 mo
	Guisier F et al., 2020 [44]	Retrospective	23/107	Second-line and later line	Mutations	NGS	ICI	Activity	ORR 27% RR 27.3% DCR 50% mDoR 15.2	mPFS: 2.2 mo OS: 20.4 mo
	Peters S et al., 2019 [45]	Phase II	49	Second-line	Overexpression	IHC breast cancer (3+ vs. 2+)	T-DM1	ORR	ORR HER3+ vs. HER2+: 20% vs. 0%	HER 3+ vs. HER2: mPFS 2.7 vs. 2.6 mo; mOS 15.3 vs. 12.2 mo
	Yang G et al., 2022 [46]	Retrospective	293	First-line	Mutations or amplification	NGS	CT vs. CT + ICI vs. CT + AI	Activity	CT: ORR 16.9% DCR 89.2% CT + ICI: ORR 28.9% DCR 80.0% CT + AI: ORR 23.8% DCR 91.3%	CT: mPFS: 4.03 mo mOS: 31.67 mo CT + ICI: mPFS: 5.20 mo CT + AI: mPFS: 5.63 mo mOS: 36.27 mo
	Song Z et al., 2022 [47]	Phase II	27	First-line and later line	Amplified	NGS	Pyrotinib	PFS	ORR 22.2%	mPFS: 6.3 mo mOS: 12.5 mo

**Abbreviations:** AI, angiogenesis inhibitors; ampl, amplified; CAP, College of American Pathologists; CI, confidence interval; CISH, chromogenic in situ hybridization; CR, complete response; CT, chemotherapy; DCR, disease control rate; Doce, docetaxel; FISH, fluorescence in situ hybridization; HNSCC, head and neck squamous cell carcinoma; ICI, immunecheckpoint inhibitor; IHC, immunostichimical; mDOR, median duration of response; mOS, median overall survival; mPFS: median progression-free survival; mTD, median treatment durantion; mo, months; MCPG, mucoepidermoid carcinoma of parotid gland; mut, mutated; N, number; NE, not estimable; NGS, next-generation sequencing; NR, not reported; ORR, overall response rate; PBO: placebo; PR, partial response; pts, patients; RR, response rate; RT-PCR, reverse transcription polymerase chain reaction; SD, stable disease; SGC, salivary gland carcinoma; SCCPG, squamous cell carcinoma of parotid gland; WT: wild-type; T-DM1, ado-trastuzumab emtansine; T-DXd, trastuzumab deruxtecan; vs., versus.

**Table 3 cancers-16-03145-t003:** Published papers with anti-HER2 agents in BTCs and CRC.

Type of Tumor	Author, Year [Ref]	Study Design	N° pts	Treatment Line	Type of HER2 Alteration Evaluated	Definition of HER2 Positivity	Drug	Primary Endpoint	Results	Survival Data
Biliary tract cancers	Javle et al., 2022 [60]	Phase II	29	Subsequent lines	Amplification/ Overexpression	IHC 2+ or 3+ (breast criteria)	Trastuzumab *plus* pertuzumab	ORR	23%	mPFS: 4.0 mo mOS: 10.9 mo
	Harding JJ et al., 2023 [61]	Phase II	25	Subsequent lines	Mutation	*HER2* gene in NGS (MSK-IMPACT)	Neratinib	ORR	16%	mPFS: 2.8 mo mOS: 5.4 mo
	Lee CK et al., 2013 [62]	Phase II	34	First-line	Amplification/ Overexpression	IHC 3+ or IHC 2+ and in situ hybridization positive or *ERBB2* gene copy number ≥ 6.0 or using NGS	FOLFOX *plus* trastuzumab	ORR	29%	mPFS: 5.1 mo mOS: 10.7 mo
	Nakamura Y et al., 2023 [63]	Phase II	30	Second- and further lines	Amplification/ overexpression	IHC 3+ or IHC 2+ and in situ hybridization positive or *ERBB2* gene copy number ≥ 6.0 or using NGS	Tucatinib *plus* trastuzumab	ORR	46%	mPFS: 5.5 mo mOS: 15.5 mo
	Meric-Bernstam F et al., 2022 [64]	Phase I	22	Subsequent lines	Amplification/ overexpression	IHC 3+ or gene amplification confirmed using FISH, according to the ASCO/CAP guidelines for gastroesophageal cancer	Zanidatamab	ORR	38%	mPFS: 3.5 mo
	Harding JJ et al., 2023 [65]	Phase IIB	80	Progression on previous gemcitabine-based therapy	Amplification/ overexpression	IHC 3+ or gene amplification confirmed by FISH, according to the ASCO/CAP guidelines for gastroesophageal cancer	Zanidatamab	ORR	41%	mPFS: 5.5 mo
	Ohba A et al., 2022 [66]	Phase II	24	Progression on previous gemcitabine-based therapy	Amplification/ overexpression	HER2-positive: IHC 3+ or gene amplification confirmed using FISH; HER2-low expression [HER2-low]: IHC/ISH status of 0/+, 1+/−, 1+/+, or 2+/−	T-DXd	ORR (HER2-positive)	36%	mPFS: 4.4 mo mOS: 7.1 mo
	Meric-Bernstam F et al., 2024 [24]	Phase II	41	Progression on ≥2 systemic treatment	Amplification/ overexpression	HER2-overexpressing tumors with IHC 3+/2+ using current ASCO/CAP guidelines for scoring HER2 in gastric cancer	T-DXd	ORR	27%	mPFS: 4.1 mo
Colorectal carcinoma	Clark et al., 2003 [67]	Phase II	21	Second or third line	Overexpression	IHC 2+ (breast criteria)	Trastuzumab *plus* FLOX	ORR	24%	mDOR: 4.5 mo (range 2.7–11 mo)
	Ramanathan et al., 2004 [68]	Phase II	9	First or second line	Overexpression/amplification	IHC 3+ or 2+ (breast criteria) confirmed by FISH	Trastuzumab *plus* irinotecan	ORR	ORR 71% PR (5)	NR
	Sartore-Bianchi et al., 2016 [69]	Phase II	27	Refractory/late lines	Overexpression and amplification	HERACLES Diagnostic Criteria	Trastuzumab *plus* lapatinib	ORR	30%	mPFS: 5.3 mo mOS: 11.5 mo
	Meric-Bernstam et al., 2019 [70]	Phase II	57	Late lines	Amplification/ overexpression and mutation	FISH/CISH, IHC and/or NGS through local testing and revaluation	Trastuzumab *plus* pertuzumab	ORR	ORR 32%, PR (17), CR (1)	mPFS: 2.9 mo estimated mOS: 11.5 mo
	Sartore-Bianchi et al., 2020 [71],	Phase II	31	Second and third lines	Overexpression/ Amplification	HERACLES Diagnostic Criteria	Pertuzumab and T-DM1	ORR	9.7%	mPFS: 4.1 mo
	Gupta et al., 2022 [72]	Phase II	28	Late lines	Amplification/ overexpression and mutation	NGS	Trastuzumab *plus* pertuzumab	ORR	14%	mPFS: 17.2 wks
	Yoshino et al., 2019 [73]	Phase II	19	Refractory	Amplification on tissue and ctDNA mutations	Evaluation on tissue (IHC and ISH) and in ctDNA using NGS (criteria not otherwise specified)	Trastuzumab *plus* pertuzumab	ORR	Tissue positive group: ORR 35%, CR (1), PR (5). ctDNA-positive group: ORR 33%, CR (1), PR (4).	mPFS: 4.0 mo
	Siena et al., 2021 [74]	Phase II	78	Third-line	Overexpression/ amplification	IHC and ISH (criteria not specified) Cohort A—53 (IHC 3+ or IHC2+ ISH-positive)	T-DXd	ORR	ORR 45.3% DCR 83.0%	mPFS 6.3 mo mOS 15.5 mo
	Strickler et al., 2023 [75]	Phase II	117	Later lines/Refractory	Overexpression/ amplification	IHC 3+, IHC 2+ (breast criteria) and FISH/CISH amplified or amplification by NGS	Tucatinib *plus* trastuzumab	ORR	ORR 38·1%, CR (3), PR (29)	mPFS: 8.2 mo mOS: 24.1 mo
	Chang et al., 2022 [76]	Phase II	16	Third line or beyond	Overexpression/ amplification	IHC (HERACLES Diagnostic Criteria), FISH or NGS	Trastuzumab *plus* pyrotinib	ORR	ORR: 50% all ORR: 57% in RAS wild type	mPFS: 7.53 mo mOS: 16.8 mo

**Abbreviations:** CAP, College of American Pathologists; CISH, chromogenic in situ hybridization; CR, complete response; ctDNA circulating tumor DNA; DCR, disease control rate; DOR, duration of response; FISH, fluorescence in situ hybridization; FOLFOX and FLOX, 5-fluorouracil, leucovorin and oxaliplatin; IHC, immunostichimical; mOS, median overall survival; mPFS: median progression-free survival; mo, months; N, number; NGS, next-generation sequencing; NR, not reported; ORR, overall response rate; PR, partial response; pts, patients; SD, stable disease; T-DM1, ado-trastuzumab emtansine; T-DXd, trastuzumab deruxtecan.

**Table 4 cancers-16-03145-t004:** Published papers with anti-HER2 agents in genitourinary and gynecological tract neoplasms.

Type of Tumor	Author, Year [Ref]	Study Design	N° pts	Treatment Line	Type of HER2 Alteration Evaluated	Definition of HER2 Positivity	Drug	Primary Endpoint	Results	Survival Data
Urothelial carcinoma	Hussain MH et al., 2008 [96]	Phase II	44	First line	Overexpression, amplification, serum HER-2/neu-ECD level	IHC 2+ or 3+ (breast criteria), serum HER-2/neu-ECD ≥ 16 ng/mL	Trastuzumab, carboplatin, paclitaxel, gemcitabine	Cardiac toxicity	22.7%	PFS: 9.3 mo (95% CI, 6.7 to 10.2 mo) OS: 14.1 mo (95% CI, 11.5 to 17.1 mo)
	Wülfing C et al., 2009 [97]	Phase II	59	Second line	Overexpression	IHC 2+ or 3+ (breast criteria)	Lapatinib	ORR > 10%	2%	TTP: 8.6 wk (95% CI, 8.0 wk to 11.3 wk) OS: 17.9 wk (95% CI, 13.1 wk to 30.3 wk)
	Galsky MD et al., 2012 [88]	Phase II	9	Second line	Amplification	FISH ratio ≥ 2	Lapatinib	12 wk-ORR	ORR 0	NR
	Oudard S et al., 2015 [98]	Phase II	61	First line	Amplification/ overexpression	IHC 3+ or gene amplification confirmed using FISH (breast criteria)	Platinum, gemcitabine ± trastuzumab	PFS	PFS 8.2 mo vs. 10.2 mo	mOS: 15.7 mo vs. 14.1 mo, (*p* = 0.684)
	Choudhury NJ et al., 2017 [99]	Phase II	23	Second and subsequent lines of therapy	Overexpression/amplification/ mutation/copy number alteration	IHC 3+ or 2+ confirmed using FISH (breast criteria)/ Mutation using NGS/ Copy number ≥ 3.5 by NGS	Afatinib	3 mo PFS	21% pts met 3 mo-PFS	mOS: 5.3 mo mPFS: 1.4 mo
	Powles T et al., 2017 [100]	Phase III	232	Second and subsequent lines of therapy	Overexpression/ Amplification	IHC 3+ and 2+ confirmed by FISH (breast criteria)	Lapatinib vs. PBO	PFS	mPFS 4.5 mo (lapatinib) (95% CI, 10.5 mo to 5.4 mo) vs. 5.1 mo (95% CI, 3.0 mo to 5.8 mo) (PBO)	OS 12.6 (95% CI, 9.0 to 16.2) and 12.0 (95% CI, 10.5 to 14.9)
	Michaelson MD, 2017 [101]	Phase I/II	66	Second and subsequent lines of therapy	Overexpression	IHC 2+ or 3+ (breast criteria)	Paclitaxel, radiotherapy ± trastuzumab	Toxicity	AEs in 35% of trastuzumab-treated pts; 30% in non trastuzumab-treated pts	NR
	Hyman DM et al., 2018 [27]	Phase II	16	Second and subsequent lines of therapy	Mutation	Mutation by NGS	Neratinib	ORR	ORR 0	mPFS: 1.8 mo
	Hainsworth et al., 2018 [102]	Phase IIa	9	Second and subsequent lines of therapy	Overexpression/ Amplification	IHC 3+ or amplification using FISH (breast criteria) Activating mutation by NGS	Trastuzumab, pertuzumab	ORR	ORR 33.3%	NR
	Banerji U et al., 2019 [103]	Phase I	16	Second and subsequent lines of therapy	Overexpression	IHC 1+, 2+, or 3+ (breast criteria)	Trastuzumab-duocarmazine	Safety and recommended dose	safe profile; recommended dose: 1·2 mg/kg	PFS 3.5 mo
	Xu Y et al., 2021 [104]	Phase I	4	Second and subsequent lines of therapy	Overexpression	IHC 2+ or 3+ (breast criteria) regardless the presence/absence of amplification by FISH	RC48-ADC	Safety and MTD	Safety profile; MTD NR	NR
	De Vries EGE et al., 2023 [105]	Phase II	13	Second and subsequent lines of therapy	Overexpression	IHC 3+ in ≥30% tumor cells	T-DM1	BOR	PR 38.5%	PFS: 2.2 mo OS: 7 mo
	Meric-Bernstam F et al., 2024 [24]	Phase II	22	Second and subsequent lines of therapy	Overexpression	IHC 2+ or 3+ (breast criteria)	T-DXd	ORR	ORR 39%	PFS: 12.8 mo OS: 7 mo
	Sheng X et al., 2024 [106]	Phase II	107	Second and subsequent lines of therapy	Overexpression/ amplification	IHC 3+ and 2+ confirmed using FISH (breast criteria)	RC48-ADC	ORR	ORR 50.5%	PFS: 5.9 mo OS: 14.2 mo
	Font A et al., 2024 [107]	Phase II	34	Second and subsequent lines of therapy	Amplification	Amplification by FISH (breast criteria)	Afatinib	6 mo-PFS	6 mo-PFS 12%	OS: 30 wk
Prostate cancer	Morris et al., 2002 [109]	Phase II	23	After androgen deprivation therapy ± radiotherapy	Overexpression	IHC 3+ and 2+ (breast criteria)	Trastuzumab ± paclitaxel	Efficacy of trastuzumab monotherapy	ORR 0	NR
	Lara PN Jr et al., 2004 [110]	Phase II	4	After androgen deprivation therapy	Overexpression/amplification	IHC 3+ and 2+ (breast cancer) confirmed using FISH (HER2 ratio > 2)	Trastuzumab or Doce.; non-responders: trastuzumab/Doce	ORR	ORR 0	PFS: 7 mo
	Ziada A et al., 2004 [111]	Phase II	18	After androgen deprivation therapy	Overexpression/amplification	IHC 3+ and 2+ confirmed using FISH (breast criteria)	Trastuzumab	Efficacy; toxicity	SD 2/18, well-tolerated therapy	NR
EnC	Lesly KK et al., 2013. [112]	Phase II	30	Second line and later lines	Protein overexpression	IHC	Lapatinib	PFS, OS	mPFS 1.82 mo	mOS: 7.33 mo
EnC + OC	Hainsworth JD et al., 2018 [102]	Phase II	7/230 (EnC) 8/230 (OC)	Second line and later lines	Amplification/overexpression/mutations	IHC, FISH, NGS	Pertuzumab *plus* trastuzumab	ORR	EnC: ORR 0% OC: ORR 13%	NR
OC	Yang Y et al., 2018. [113]	Retrospective	80	Second line and later lines	NR	NR	Trastuzumab vs. trastuzumab *plus* abraxane	PR	PR 44.2% vs. 45.9% mo	OS 7%vs 7.3% mo
EnC, serous Histotype	Fader AN et al., 2018 [114]	Phase II	61	Second line and later lines	Protein overexpression and amplification	IHC + FISH	Carboplatin and paclitaxel +/− trastuzumab	mPFS	mPFS: 12.9 mo vs. 8.0 mo	See results
EnC + OC + CeC	Destiny-Pan Tumor Trial [115]	Phase II	ongoing	Second line and later lines	Protein overexpression	IHC	T-DXd	ORR	ORR EnC: 85% ORR CeC: 75% ORR OC: 45%	NR
EnC	Moustapha H et al., 2021 [116]	Phase II	28	Second line and later lines	Protein overexpression/amplifications/ mutations	NR	Pertuzumab *plus* trastuzumab	ORR	DCR: 37% ORR 7.1%	OS: 53.4% (1 year) mPFS 28.1 wks
EnC + OC + CeC	Meric-Bernstam F et al., 2024 [24]	Phase II	EnC 40 out 267; OC 40 out 267; CeC 40 out 267	Third line and later lines	Protein overexpression	IHC	T-DXd	ORR	EnC: 84.6% CeC 75% OC 63.6%	EnC: PFS: 11.1 mo CeC and OC: NS

**Abbreviations:** ADC, antibody–drug conjugates; BOR, best overall response; CAP, College of American Pathologists; CeC, Cervical cancer; CI, confidence interval; Doce, docetaxel; EnC, Endometrial Cancer; FISH, fluorescence in situ hybridization; HT, hormone therapy; HR, hazard ratio; IHC, immunohistochemical; mOS, median overall survival; mPFS: median progression-free survival; mTD, median treatment duration; mo, months; MTD, most tolerated dose; N, number; NGS, next-generation sequencing; NR, not reported; NS, not significative; OC, Ovarian Cancer; ORR, overall response rate; PBO: placebo; PR, partial response; pts, patients; SD, stable disease; T-DM1, ado-trastuzumab emtansine; T-DXd, trastuzumab deruxtecan; TTP, time to progression; wk, week.

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
