# Peer review of "Prognostic and Predictive Roles of HER2 Status in Non-Breast and Non-Gastroesophageal Carcinomas"

_cancers, 2024, doi:10.3390/cancers16183145_

Round 1

Reviewer 1 Report

Comments and Suggestions for Authors

I consider this to be an extremely interesting manuscript on the topic it addresses, with great relevance in oncology.

However, the way in which the topic is addressed is complex, with a reading that overwhelms the reader. The tables contain too much information from which it is difficult to reach a conclusion.

I suggest that when addressing each type of cancer, the approach be divided into relevant topics, % of positivity, categorization system used, clinical trials, prognostic significance or clinical significance with treatment, etc. Choose the necessary topics to develop, divided with subtitles, but I suggest that these topics be homogeneous for all cancers. In this way, whoever reads knows exactly where to find the topic of interest. This is because in the current state there are very long paragraphs and it is not known if one will find what one is looking for. If there is no information on a particular topic for a cancer, then say so.

The tables are very extensive, I suggest dividing it into several sections

Other strategies can be used to make the reading of the manuscript more efficient and organized, but in its current form it is very complex to read.

Author Response

Comments to the Author: I consider this to be an extremely interesting manuscript on the topic it addresses, with great relevance in oncology.

However, the way in which the topic is addressed is complex, with a reading that overwhelms the reader. The tables contain too much information from which it is difficult to reach a conclusion.

Answer: We thank the reviewer for his/her comments. We attempted to satisfy all requests.

1. Comment: I suggest that when addressing each type of cancer, the approach be divided into relevant topics, % of positivity, categorization system used, clinical trials, prognostic significance or clinical significance with treatment, etc. Choose the necessary topics to develop, divided with subtitles, but I suggest that these topics be homogeneous for all cancers. In this way, whoever reads knows exactly where to find the topic of interest. This is because in the current state there are very long paragraphs and it is not known if one will find what one is looking for. If there is no information on a particular topic for a cancer, then say so.

Answer: We thank the reviewer for the suggestions. We divided all paragraphs with subtitles as suggested and the topics are now homogeneous for all cancers. Furthermore, we have clarified the prognostic and predictive role of HER2 alterations for each neoplasm, as well as the diagnostic criteria used to evaluate HER2 alterations and the possible clinicopathological associations with HER2 alterations.  

2. Comment: The tables are very extensive, I suggest dividing it into several sections

Answer: We thank the reviewer for this comment. As you suggested, we reorganized the tables and deleted the less informative data for the purpose of the review. 

3.Comment: Other strategies can be used to make the reading of the manuscript more efficient and organized, but in its current form it is very complex to read.

Answer: We thank the reviewer for this suggestion. We are aware of the complexity and density of information contained in the manuscript. However, we have reorganized the entire manuscript so that now it is more simple and easier to read.

Reviewer 2 Report

Comments and Suggestions for Authors

The manuscript reviews status of HER2 oncogene in malignancies other than breast cancer. HER2 amplification was connected with worse prognosis but now there is targeted therapy with antibody which extended the survival of the patients very much. Now Herceptin was conjugated with deruxtecan what improved the survival of patients even more.

The authors review the most frequent cancers, describing the percentage of tumors with the overexpression of HER2 where patients could benefit or not from ADC therapy with Herceptin. Overexpression of HER2 in other than breast cancers is also connected with more aggressive phenotype. The authors show that HER2 testing is still necessary for some cancer types and that not all patients showing HER2 amplification would benefit from Herceptin therapy.

The manuscript presents very much information about testing and treating the patients according to the HER2 status.

There is problem with the numbering of pages.

Author Response

Comments to the Author: The manuscript reviews status of HER2 oncogene in malignancies other than breast cancer. HER2 amplification was connected with worse prognosis but now there is targeted therapy with antibody which extended the survival of the patients very much. Now Herceptin was conjugated with deruxtecan what improved the survival of patients even more.

The authors review the most frequent cancers, describing the percentage of tumors with the overexpression of HER2 where patients could benefit or not from ADC therapy with Herceptin. Overexpression of HER2 in other than breast cancers is also connected with more aggressive phenotype. The authors show that HER2 testing is still necessary for some cancer types and that not all patients showing HER2 amplification would benefit from Herceptin therapy.

Answer: we thank the Reviewer for her/his comment. According to Reviewer’s comment, we have added a sentence in the conclusion paragraph to emphasize and clarify that HER2 positive tumors are generally associated with a poor prognosis due to the biological role of HER2 but, in some cases, treatments have improved patient outcomes.

- Comment: The manuscript presents very much information about testing and treating the patients according to the HER2 status.

Answer: We are aware of the complexity and density of information contained in the manuscript, but with this review, we have aimed to be as comprehensive as possible on the topic. However, we have reorganized the entire manuscript adding subtitles to each paragraph so that now it is more simple and easier to read.

- Comment: There is problem with the numbering of pages.

Answer: We thank the reviewer for this comment. We corrected the manuscript’s page numbering.

Round 2

Reviewer 1 Report

Comments and Suggestions for Authors

The authors have substantially improved the manuscript, responding appropriately to the suggestions. I therefore suggest publishing the manuscript.